# An Overview of the Western United States Dynamically Downscaled Dataset (WUS-D3)

[1]Stefan Rahimi, [1]Lei Huang, [1]Jesse Norris, [1]Alex Hall, [1]Naomi Goldenson, [1]Will Krantz, [1]Benjamin Bass, [1]Chad Thackeray, [1]Henry Lin, [1]Di Chen, [1]Eli Dennis, [2]Ethan Collins, [3]Zachary J. Lebo, [1]Emily Slinskey, and the UCLA Center for Climate Science Team[+]

[1]Center for Climate Science, University of California Los Angeles, Los Angeles, California, 90095, U.S.A.
[2]Department of Atmospheric Science, University of Wyoming, Laramie, Wyoming, 82071, U.S.A.
[3]School of Meteorology, University of Oklahoma, Norman, Oklahoma 73019, U.S.A.
[+]A full list of Authors appears at the end of this manuscript

*Correspondence to*: Stefan Rahimi (s.rahimi@ucla.edu)

**Abstract.** Predicting future climate change over a region of complex terrain, such as the western United States (U.S.), remains challenging due to the low resolution of global climate models (GCMs). Yet climate extremes of recent years in this region, such as floods, wildfires, and drought, are likely to intensify further as climate warms, underscoring the need for high-quality predictions. Here, we present an ensemble of dynamically downscaled simulations over the western U.S. from 1980–2100 at 9-km grid spacing, driven by sixteen latest-generation GCMs. This dataset is titled the Western U.S. Dynamically Downscaled Dataset (WUS-D3).

We describe the challenges of producing WUS-D3, including GCM selection and technical issues, and we evaluate the simulations' realism by comparing historical results to temperature and precipitation observations. The future downscaled climate change signals are shaped in physically credible ways by the regional model's more realistic coastlines and topography: (1) The mean warming signals are heavily influenced by more realistic snowpack. (2) Mean precipitation changes are often consistent with wetting on the windward side of mountain complexes, as warmer, moister air masses are uplifted orographically during precipitation events. (3) There are large fractional precipitation increases on the lee side of mountain complexes, leading to potentially significant changes in water resources and ecology in these arid landscapes. (4) Increases in precipitation extremes are generally larger than in the GCMs, driven intensified local atmospheric updrafts tied to topography. (5) Changes in temperature extremes are different from what is expected by a shift in mean temperature and are shaped by local atmospheric dynamics and land surface feedbacks. Because of its high resolution, comprehensiveness, and representation of relevant physical processes, this dataset presents a unique opportunity to evaluate societally relevant future changes in western U.S. climate.

## 1 Introduction

Predicting climate change on a regional level is critical for assessing its societal impacts, such as changes to water resources, flooding, drought, heat waves, wildfire, and windstorms. Current-generation global climate models (GCMs) are ill-equipped for this task due to their coarse grid spacing (on the order of 1 degree longitude/latitude). This prevent GCMs from representing complex terrain and from resolving small-scale meteorological phenomena that define local hydroclimate. To counter this limitation, a regional climate model (RCM) may be used to dynamically downscale the GCM projections over a limited area. The resulting high-resolution output allows us to study future weather- and climate-relevant processes that may unfold across a region of complex terrain and gain physical insights into the land–atmosphere drivers of regional climate change. Moreover, the output can be used to drive land-surface, hydrological, and fire models under future climate conditions.

The western United States (WUS) is a particular region of interest for such heterogeneous patterns of historical climate and future climate change. It consists of major mountain ranges, deserts, shrublands, temperate forests, plains, and a complex coastline. It is affected by diverse atmospheric phenomena, such as extratropical cyclones, atmospheric rivers, persistent blocking highs, the North American Monsoon, summertime convective storms, wildfire-related downslope winds, and cooling coastal breezes. The complex interplay of these phenomena with local topography makes it impossible for GCMs to represent the diversity of microclimates within the WUS and how they may uniquely respond to larger-scale climate change. In general, GCMs project mid-latitude wetting to the north of the region and subtropical drying to the south, but with disagreement on where within the WUS the transition occurs (Meehl et al., 2007; Neelin et al., 2013). Moreover, intensified interannual swings

between extremely wet and extremely dry years (i.e., 'whiplash') are projected in parts of the region (Swain et al., 2018; Chen et al., 2022). In recent years the WUS has experienced catastrophic weather/climate events, such as the southwestern U.S. drought (Mankin et al., 2019.; White et al., 2023), record-breaking floods in California in 2017 (White et al., 2019) and 2023, and the unprecedented 2021 heatwave in the Pacific Northwest (White et al., 2023). In a warming climate, all of these extreme events are likely to be intensified. Thus, dynamical downscaling of future GCM projections over the WUS can provide a unique insight into how large-scale climate change may interact with its complex terrain and diverse meteorological phenomena.

Direct dynamical downscaling of GCMs is far less common than that driven by historical reanalyses (Liu et al., 2017, 2011; Rahimi et al., 2022; Rasmussen et al., 2011, 2014; Norris et al., 2019, and many, many others), despite the fact that it yields a representation of local weather and climate based on atmospheric and land surface physics (Bruyère et al., 2014; Coppola et al., 2020, 2021; Huang et al., 2020, 2021; Komurcu et al., 2018; Wang and Kotamarthi, 2015, 2013; Zobel et al., 2018, 2017; Bukovsky and Karoly, 2011; Bukovsky et al., 2021; Mearns et al., 2012; Scalzitti et al., 2016). In this respect, it is superior to other downscaling methods. For example, because it relies on physics for its predictions, dynamical downscaling of GCMs does not assume stationarity (Lanzante et al., 2018) in its future projections, as with other forms of downscaling (e.g., statistical). Hence, it can produce plausible extreme weather events that cannot be found in the historical record and may only emerge in an unobserved future climate. Additionally, RCMs can solve for the full complement of physical quantities relevant to climate. These variables are not typically obtained through statistical methods or via machine learning techniques. Such techniques typically focus on a small set of variables, due to missing

high-resolution observational data products and hence training data. For example, statistically downscaled precipitation and temperature data products, even when obtained using multivariate relationships, may contain no information about water vapor content, surface pressure, cloud depth, etc. Finally, the use of physics to arrive at the downscaled result means that feedbacks between the landscape and the overlying atmosphere, and other land and atmosphere processes, may be effectively simulated.

There are three significant barriers to using RCMs to dynamically downscale GCMs: (1) RCMs require sub-daily three-dimensional variables as initial and boundary conditions, which are not typically sufficiently archived in GCM databases; (2) RCM configurations may not be designed to ingest GCM data as boundary conditions; and (3) It is extremely computationally expensive.  Because of these barriers, dynamical downscaling of full GCM ensembles (e.g., the Coupled Model Intercomparison Project Phase 6; CMIP6) at landscape-resolving (~10 km) grid spacings generally remain out of reach.

Despite these barriers, we present results from sixteen new dynamically downscaled CMIP6 simulations over 11 WUS states, including the whole of the Western Electricity Coordinating Council (WECC) region, comprising the Western U.S. Dynamically Downscaled Ensemble (WUS-D3). These simulations span 1980-2100, combining the historical and Shared Socioeconomic Pathway (SSP) output for each GCM. Downscaling a wide variety of CMIP6 models yields a diverse suite of possible future climates over the WUS at a landscape-resolving scale (9-km grid spacing). In the following sections, we present our methodology and technical challenges encountered, as well as a characterization of the historical performance and future change signals from our dataset.

## 2 Methodology

### 2.1 WRF Setup

We use the Weather Research and Forecasting (WRF) model version 4.1.3 (Skamarock et al., 2019) to dynamically downscale the simulations of 14 CMIP6 GCMs (Table 1) from 1980–2100. In each simulation, historical forcing was applied up to 2014, and then the forcing associated with the SSP3-7.0 scenario thereafter. SSP3-7.0 is a high-emissions scenario in which greenhouse-gas emissions double by end-of-century (O'Neill et al., 2016). We also downscale one GCM's (CESM2) SSP-2-4.5 and SSP-5-8.5 projections. In these scenarios, emissions remain roughly constant until 2050 before falling thereafter, and triple by end-of-century, respectively.

WRF is configured as was documented for a related downscaling of the ECMWF Fifth Generation Reanalysis (ERA5; Hersbach et al., 2020) in Rahimi et al. (2022; WRF-ERA5). We downscale each GCM year separately and in parallel; at the beginning of each downscaling period (on August 1), the RCM is initialized to the driving GCM state. In this way, an *N*-year simulation can be completed in the same wall clock time as a 1-year experiment. For each year of integration, we choose the beginning of the retained WRF output for analysis to coincide with minimum snowpack across the WUS (September 1). This approach produces one month of spin-up for the land surface. Thus, WRF is initialized on August 1 to surface and three-dimensional data from each GCM and integrated through September 1 of the following year (13 months, including the spin-up month) on 39 atmospheric levels. This approach is similar to that of Zobel et al. (2018, 2017), who also initialized WRF experiments at yearly intervals, but only included 1 day of model spin-up. Despite our 1-month spin-up, soil moisture,

land surface fluxes, and streamflow may still suffer from minor biases due to imperfect soil texture categories and their associated hydrophysical properties (Dennis and Berbery, 2021). However, because soil texture is a necessary component of the land surface model, and these underlying datasets are imperfect, these effects are somewhat unavoidable without massive regional calibration. WRF's parallelization procedure, which is advantageous for executing simulations in weeks instead of years, is performed to the detriment of time continuity in simulating the surface and subsurface runoff with high precision. To address this issue, we propose that the atmospheric fields from WRF be used to drive offline and calibrated hydrology models that are continuous (e.g., Bass et al., 2023).

Atmospheric carbon dioxide and methane concentrations vary yearly in our simulations based on northern-hemispheric-mean values from input4MIPs (Durack et al., 2017). Prior to 2015, CMIP6 historical values are prescribed. From 2015 onward, these values are taken from the SSP3-7.0 scenario, except for the alternate SSP CESM2 experiments. WRF's radiative code is modified to enable concentrations to be manually inputted; this modification is no longer needed as of WRF version 4.4.2. Historical 21-category land-use/land-coverage information from the Moderate Resolution Imaging Spectrometer is used in all experiments.

We dynamically downscale all GCMs to two grids of 45-km and 9-km grid spacing (Figure 1). On the parent 45-km grid, the horizontal winds, temperature, and geopotential height are relaxed (relaxation coefficient of 0.0003 $s^{-1}$) to their respective GCM-simulated fields above the planetary boundary layer via spectral nudging for wavelengths greater than 1,500 km. Smaller waveforms are allowed to evolve freely on the WRF grid (Spero et al., 2014). This approach is designed to reduce internal model drift away from the GCM state. One-way nesting is then used to dynamically downscale

the 45-km result to the 9-km grid, on which spectral nudging is not implemented. The 9-km grid encompasses the entirety of the WECC's U.S. coverage area.

The lateral boundary conditions are updated at 6-hourly intervals, and adaptive time stepping is used. Convective precipitation is parameterized following Tiedtke (1989) and Zhang et al. (2011). P3 microphysics is used (Morrison and Milbrandt, 2015), shortwave and longwave radiation schemes of Iacono et al. (2008) are implemented, and the Noah land surface model with multi-parameterizations (Noah-MP) is used (Niu et al., 2011).

**2.2 GCM Selection**

Prioritizing SSP3-7.0 with an end-of-century radiative forcing of 7 W $m^{-2}$, we selected 14 GCMs (Table 1) based on three criteria: (i) their skill in simulating important processes that govern western North American climate over the historical (1980-2010) period, (ii) their collective representativeness of the broader CMIP6 ensemble spread in future temperature and precipitation responses, and (iii) data availability. Processes considered in the GCM evaluation included:

1. Large-scale meteorology associated with Santa Ana and Diablo winds
2. The El Niño Southern Oscillation (ENSO)
3. Northern Hemisphere blocking (Simpson et al., 2020)
4. Landfalling jet characteristics
5. GCM-simulated temperature and precipitation
6. Extreme precipitation across California

This ranking system is described in Krantz et al. (2021), and the process of choosing GCMs to downscale based on end-user needs and locally relevant atmospheric processes is described in Goldenson et al. (2023; in revisions). In addition, being subject to these selection processes, the GCMs downscaled in this study span the spread in the future changes in temperature and precipitation from CMIP6 across the WUS.

We only dynamically downscale GCMs with the following outputs archived on the Earth System Grid Federation system: 3-D atmospheric temperature (ta), horizontal winds (ua and va), and specific humidity (hus), surface pressure (ps), soil layer-specific temperature and water content (tsl and mrsol, respectively), and sea surface temperature (SST; tos). Furthermore, we only dynamically downscale GCMs with 6-hourly instantaneous atmospheric outputs defined on native model levels ("6hrLev") rather than on isobaric surfaces ("6hrPlevPt"). Generally, 6hrPlevPt GCM outputs are only defined on 3-10 pressure surfaces which may be problematic for atmospheric phenomena characterized by more granular vertical structures. In testing, we found that this vertical resolution can have a large impact on the downscaled solution.

Additionally, we require that the full time series of SSTs be available in GCM outputs. These SSTs are then prescribed in WRF and update daily, which may be problematic for atmospheric processes subject to a strong atmospheric-ocean coupling and evolve over sub-daily time scales. To bypass this issue, we tested using a slab ocean model in WRF. With time, strange artifacts in the SST and outgoing longwave radiation fields gradually developed, so slab ocean physics were not enabled and its use discourage for simulations on regional climate time scales (Ming Chen, personal communication; https://forum.mmm.ucar.edu/threads/weird-pixilated-skin-temperatures-when-using-

sf_ocean_physics-1.12693/). Daily SSTs are available for most GCMs, except for FGOALS-g3 and GISS-E2-1-G, which only made monthly SST outputs available. Thus, in the cases of FGOALS-g3 and GISS-E2-1-G, linear interpolation is used to upsample monthly mean SSTs (assumed to be valid at the midpoint of each month) to daily values.

### 2.3. Sea surface temperatures in the Gulf of California

SSTs in the Gulf of California (GoC) are known to modulate the North American Monsoon, which provides roughly a third of Arizona and New Mexico's annual precipitation (Mitchell et al., 2002). However, the GoC is poorly resolved, if at all in CMIP6 GCMs; in the best case, the GoC is expressed as a subtle bay that barely intrudes into the North American continent. As a result, there is generally no SST information from GCMs across the GoC that can be used to directly prescribe SSTs in the WRF-resolved GoC. An additional problem is that the adjacent open Pacific SSTs are on average about 10 K lower and undergo less seasonal variability than in the GoC (Figure 2). Hence, linearly extrapolating from the adjacent open Pacific to the GoC would produce a representation of GoC SSTs that is clearly unphysical. Fortunately, there are predictable relationships in ERA5 between the climatological GoC entrance region temperature (Fig. 2c), which can be taken directly from GCMs, and the along-axis GoC SST gradient (Fig. 2d), which can be used to produce reasonable SSTs within the GoC. Thus, in most GCMs, we apply the following linear extrapolation to estimate GoC SSTs based on the entrance-region SSTs:

$$T_{GoC} = -\left.\frac{\partial T}{\partial n}\right]_{ERA5} (n) + T_{entry,GCM} \qquad (1)$$

where $\left.\frac{\partial T}{\partial n}\right]_{ERA5}$ is the monthly varying climatological GoC temperature gradient from ERA5 and is always positive, $n$ is the along-GoC axis coordinate (pointing towards the northwest), and $T_{entry,GCM}$ is the GoC entrance temperature, which is resolved in GCMs. The relevant regions are outlined in Figure 2a. To our knowledge, the difficulty in dealing with SSTs in coastal estuaries and gulfs has not been generally addressed in regional climate modeling efforts, and this is the first time that a physically based mathematical relationship has been used to address this issue across this region.

We apply the above linear extrapolation to all GCMs, except CESM2, CNRM-ESM2-1, and MPI-ESM1-2-LR, which were all downscaled prior to implementation of this improvement. Consequently, for CESM2 and MPI-ESM1-2-LR, there is a spurious SST discontinuity (Figure S1). This is due to the default extrapolation routine used in WRF, which uses a nearest weighted gridpoint averaging approach to prescribe GoC SSTs. Thus, in the southern GoC, the default extrapolation uses the nearest GCM grid points from the warm GoC entrance region, whereas further north the closest GCM ocean grid cells are (inappropriately) from the Open Pacific. The discontinuity and unrealistically low SSTs in the northern GoC in these simulations may affect the simulation of the North American Monsoon but are unlikely to affect other WUS phenomena documented in this paper. For CNRM-ESM2-1, we masked out the southern GoC to remove this discontinuity in extrapolation, leading to its SSTs being homogeneously populated by Open Pacific SSTs. While smooth (Fig. S1), this approximation is less physical than the improvement described above (Fig. 2).

### 2.4 Interpolation strategy

WRF requires all atmospheric, land, and ocean GCM inputs to be defined on a rectilinear grid with atmospheric variables defined on isobaric surfaces. However, some GCMs' outputs are given on irregular atmospheric grids, whose latitude coordinates are not equally spaced from pole to pole. FGOALS-g3 for instance is characterized by ~5° latitudinal grid spacing near the poles and ~2° grid spacing near the equator. Thus, for the GCMs without a native rectilinear grid, we interpolate the output to rectilinear grids with grid spacings defined by their respective absolute minimum latitude or longitude grid spacing. This technique preserves the smallest-scale features resolved on the native GCM grid.

Since GCMs use different land surface models (LSMs) containing differently defined vertical coordinates, we generally interpolate volumetric soil moisture and soil temperature from the native LSM levels to 3.5, 14, 64, and 195 cm. In instances where vertical interpolation was not used, we used the GCM's native grid soil information. Volumetric soil moisture was computed using the CMIP6 variable, mrsol, the layer total water content, and dividing it by the layer thickness and the density of water. GCM soil fields were generally available daily.

## 2.5 Other Technical Challenges

In this section, we present additional technical challenges and known issues in the downscaled data. First, WRF is not designed to ingest GCM inputs that are, depending on the modeling center, defined on different vertical coordinates. For instance, CESM2 uses a hybrid-pressure, FGOALS-g3 a sigma, and UKESM1-0-LL a hybrid-height vertical coordinate system. As a result, unique routines had to be developed for each GCM to convert their model level output to WRF-usable inputs on isobaric

pressure surfaces. This prevented the development of a one-size-fits all routine to preprocess GCM outputs for ingestion by WRF. This issue was compounded by the fact that some GCMs, such as UKESM1-0-LL and ACCESS-CM2, contain staggered outputs on their native Arakawa C-grids.

Second, 6hrLev GCM atmospheric fields are generally provided at 0000, 0600, 1200, and 1800 UTC. However, for the entire FGOALS-g3 and historical (1980-2014) component of the NorESM2-MM experiments, data were provided at 0300, 0900, 1500, and 2100 UTC. For FGOALS-g3, we simply integrated all experiments from 1 August 1980 0300 UTC through 1 September 2100 0300 UTC. For NorESM2-MM however, we linearly interpolated the historical GCM data to 0000, 0600, 1200, and 1800 UTC before downscaling. As a further aside, since UKESM1-0-LL uses a 360-day calendar, we had to modify WRF's source code accordingly. WRF is designed by default to function with Proleptic Gregorian calendars (e.g., ERA5, MPI-ESM1-2-LR, EC-Earth3-Veg), but we compiled the model with no-leap calendars for other GCM experiments (e.g., CESM2, GISS-E2-1-G, TaiESM1).

## 3. Simulation of the Historical Climate

Next, we present a review of simulated historical (1981-2010) precipitation and surface air temperature across the WUS. Figure 3 shows the added value introduced by dynamical downscaling in simulating these patterns, as well as the relative fidelity of the GCMs when downscaled with WRF. We compare the downscaled ensemble mean against the native-resolution GCM ensemble mean, in addition to WRF-ERA5 and observational estimates from the Parameter-elevation Regressions on Independent Slopes Model (PRISM; Daly et al., 1994). The inability of the raw GCMs to capture the complex

terrain of the WUS is illustrated by major warm and cold biases over mountains and valleys, respectively. In particular, California's Central Valley is 5-7 K too cool, while the Sierra Nevada is warm-biased by the same magnitude. By contrast, the dynamically downscaled simulations, whether GCM- or ERA5-driven, better resemble the regional temperature and precipitation patterns shown by PRISM. Despite this improvement, the downscaled GCM experiments are generally colder than PRISM, by as much as 5 K during part of the year in some states (Figure 4). The annual-mean spatial patterns in Fig. 3 reveal the cold biases to be predominantly over mountains. The cold bias is generally most prominent in the winter months (shown by spatial patterns in Figure S2) but persists year-round. Additionally, dynamical downscaling generally reduces the simulated temperature spread from that of the parent GCMs, as indicated by the black circles (Fig. 4). Exceptions are noted across some western states, especially in winter.

The dynamically downscaled ensemble mean is generally too wet across the states of Washington, Oregon, and California (Figure 3; Figure 4). A preexisting wet bias in the parent GCMs is increased by downscaling, an impact seen primarily over mountains during winter (Figure S3). These biases vary substantially within the ensemble, with individual downscaled GCMs exhibiting meaningful subregional biases of hundreds of percent (e.g., in May for CNRM-ESM2-1; not shown). However, the downscaled results greatly improve on large wet biases across Nevada, Colorado, Wyoming, and Montana in the parent GCMs, which are as much as 50% in the ensemble mean across Wyoming, and hundreds of percent in some GCMs. Also, across Arizona, the summertime precipitation maximum is completely missed in all GCMs. Meanwhile, the downscaled results capture it well, albeit with some simulations far too wet (~100% bias) compared to PRISM. Difficulties in simulating summertime

precipitation across the southwestern U.S. have been noted in previous studies (Liu et al., 2017; Rahimi et al., 2022). Dynamical downscaling generally increases the simulated precipitation spread across Arizona, California, Oregon, and Washington whilst decreasing the spread across interior states.

In general, overly wet and cold dynamically downscaled GCMs have previously been noted across the region with a different RCM (Rastogi et al., 2022), indicating that biases in the GCM forcing data may be responsible and has been explored in Rahimi et al., (2023; submitted). We largely do not find such large biases in WRF-ERA5 (shown for spatial patterns in Figure 3 and seasonal cycles in Figure 4), which is equivalent to the downscaled GCMs, except driven by ERA5.

Finally, we evaluate historical extreme precipitation (rx1day) across the WUS in the dynamically downscaled ensemble. Dynamical downscaling markedly improves the spatial distribution of rx1day across the region compared to the parent GCMs (Figure 5, top), as with mean precipitation (Fig. 3). Across individual states, dynamical downscaling produces rx1day magnitudes that are in many cases about double their parent GCM values, particularly across Arizona, California, Oregon, and Washington. While generally too wet compared to PRISM, downscaled simulations are much closer to the downscaled reanalysis (WRF-ERA5). We attribute the greater rx1day values in the downscaled simulations to the much better representation of topography and orographic precipitation in WRF compared to the parent GCMs. As such, the wetter behavior of WRF solutions is generally localized to the highest elevations across each state. These locations are precisely where observational uncertainties are also maximized (Lundquist et al., 2019). Thus, we characterize downscaled rx1day simulations as being wetter than PRISM, rather than clearly being wet biased. Because of the rareness of rx1day

events, the computation of rx1day is also sensitive to the phasing of internal climate variability, which is different in GCMs relative to PRISM and WRF-ERA5.

As indicated by the shaded bars in Figure 5, downscaling may alter the original GCM spread in simulated rx1day magnitudes. Specifically, WRF significantly increases the GCM spread in Oregon, Arizona, New Mexico, and Colorado, but significantly decreases the spread in California, Nevada, Idaho, and Montana. The states with large increases in spread are generally where rx1day is more likely to occur during summer, indicating disagreements in monsoon-related extreme precipitation across downscaled results. The amplification of model uncertainty in precipitation extremes by dynamical downscaling is yet to be addressed by the regional modeling community and is a current focus of our research efforts.

## 4. Climate Response Across the Western U.S.

Next, we provide an overview of the dynamically downscaled ensemble's climate response to anthropogenic forcing (following SSP3-7.0). Figure 6 shows the mid-century (2030–2060; MC) and end-century (2070–2100; EC) projected changes in annual-mean precipitation scattered against warming, averaged across 11 WUS states in each GCM. The native GCM projections (indicated by letters) are connected to their downscaled counterparts (indicated by circles) by thick arrows.

According to the downscaled ensemble, the WUS will experience 2.25 ±0.58 K of warming by MC, and 4.65 ± 1.14 K by EC (relative to 1980-2010). A considerably more uncertain but generally wetter future is also predicted, with an ensemble mean precipitation change of 0.039 ± 0.93 mm $d^{-1}$ by MC and 0.083 ± 0.13 mm $d^{-1}$ by EC. Despite a positive mean change, a handful of simulations suggest

drying across the region. Downscaling generally preserves the inter-model variation among the parent GCMs in the 11-state mean. For warming amounts, there are correlation coefficients of 0.96 and 0.98 for MC and EC, respectively, between the raw GCM and downscaled ensembles. Correlation coefficients are lower for precipitation change but remain high: 0.88 and 0.78 for MC and EC, respectively. Regional-mean GCM warming is typically modified by no more than 0.5 K. Interestingly, downscaling generally reduces warming (leftward pointing arrows). This effect is most prominent during winter and spring (Figure S4), indicating that much better resolution of topography and hence climatological snowpack improvements (e.g., Walton et al., 2017) in the downscaling may be reducing the overall snow albedo feedback and hence the surface's temperature sensitivity to anthropogenic forcing. Summer will see the largest mean temperature increases across the WUS by EC, 5.2 ± 1.2 K in the WRF simulations compared to 5.3 ± 1.2 K in the GCMs. In contrast to temperature, resolution of topography does alter the precipitation signals significantly, but not in any systematic or obviously predictable way; downscaling can either wetten or dry the GCM precipitation projection. These modifications are generally no more than 0.05 mm $d^{-1}$, but notably CanESM5 and FGOALS-g3's projections are altered by –0.2 mm $d^{-1}$ and +0.15 mm $d^{-1}$, respectively by EC. In the case of CanESM5, this transforms strong wetting to weak drying.

### 4.1 Spatial Patters of Temperature and Precipitation Change in WRF versus GCMs

Although domain-mean changes are minimally unaffected by downscaling, the spatial patterns of temperature and precipitation change in the downscaled solutions are significantly different from those of the raw GCM projections (Figure 7, individual downscaled GCM annual changes are shown in

Figs. S5, S6). To account for large intermodel spread in climate sensitivity, the local warming is normalized by EC changes in global warming. A value of 2 K $K^{-1}$ indicates that a gridcell warms at twice the rate of the global average. Examining the upper panels of Figure 7, large-scale spatial patterns of warming are preserved in the downscaling, but there are seasonal and local differences. Notably, we see enhanced (and likely more realistic) warming adjacent to mountainous areas of the Rockies during winter and spring and at the highest elevations of the Sierra Nevada during summer. This is primarily tied to the improved representation of topography, and thus more expansive historical snow cover. We expect simulations with greater snow cover to exhibit more warming across these areas because they would be able to lose more snow under warming (Figure S7) and therefore have a stronger snow albedo feedback (SAF: Hall 2004; Qu and Hall 2006; Thackeray and Fletcher 2016). The addition of value is clear in terms of the granularity of future snow loss and subsequent impact on warming in winter and spring and is comparable to previous studies (e.g., Walton et al. 2017). However, enhanced summertime high-elevation warming may be somewhat overestimated due to large cold-season wet biases (Figures 4, S3); excessive snow survives into the warm season and creates an unrealistically large snow albedo feedback effect under climate change. We also hypothesize that the lapse rate feedback (Hansen et al., 1984, Colman and Soden, 2021) may be contributing to the enhanced warming at high elevations during summertime. For example, in the GCMs 850 hPa temperatures warm by 3.75 K across the WUS, while 300 hPa warm by 4.75 K (not shown). This enhanced warming at high altitudes likely contributes to enhanced surface warming at high elevations as well. Lastly, the downscaled ensemble exhibits enhanced warming across the interior during fall, perhaps associated with drying and a reduction in evaporative damping of surface temperature (Zhou et al., 2019).

Spatial patterns of precipitation change illustrate greater contrasts between the downscaling and GCMs across seasons (Figure 7, lower panels). EC changes are normalized by global-mean warming here to take into account the large spread in climate sensitivity among CMIP6 models, and to focus attention on the components of the hydrologic response that do not simply scale with temperature. The large-scale precipitation response is generally preserved in downscaling, with statistically significant wetting (drying) in the northwestern (southwestern) U.S. during winter and spring. The lack of statistical significance along a transition region, extending from southern California through northern Arizona and New Mexico, is symptomatic of GCM disagreement on the location of transition of subtropical drying to mid-latitude wetting (Meehl et al., 2007; Neelin et al., 2013). Consistent with other studies (e.g., Mahoney et al. 2021; Rupp et al. 2022), the downscaled ensemble appears to produce greater wetting across major WUS mountain ranges during spring and winter. The locally more intense change signal is tied to increased water vapor within atmospheric rivers and other synoptic disturbances, which interacts with WRF's more realistically simulated terrain to produce more realistic orographic uplift relative to native-resolution GCMs (Huang et al., 2020; see mean changes in vertical velocities; Fig. S8). There are also instances where WRF simulates a locally more intense drying signal compared to the native GCMs, which is also clearly linked to topography, e.g., the Sierra Nevada in autumn and spring, the upslope of the Cascades in summer, and northwestern Mexico in winter and summer.

We also examine ensemble-mean fractional precipitation changes (again normalized by warming), to focus attention on where the largest changes are relative to the climatology (Figure 8). One of the most notable and robust signals, seen during all seasons and almost entirely missed in the

parent GCMs (Fig. S9), is significant wetting in the lee of major WUS mountain ranges. This effect was explored in an idealized context in Siler and Roe (2014). They concluded that higher cloud bases associated with decreased surface relative humidity values in a warmer world will lead to enhanced hydrometeor fallout further upslope and downwind of mountain ranges. In our simulations, these lee-side changes are large in magnitude. For example, in winter, precipitation increases by 7-10% $K^{-1}$ in the lee of the Cascades, 10-20% $K^{-1}$ in the lee of the Sierra Nevada, 6-20 % $K^{-1}$ over California's Central Valley (i.e., the lee of the coastal ranges), and otherwise 5-20% $K^{-1}$ in lee-side watersheds of the intermountain West, including the entire western Great Plains. In spring, this lee-side wetting response is limited to northern mountain ranges such as the Cascades, Wyoming ranges, and the Northern Great Plains. During summer, the downscaling also shows a dipole of drying (wetting) over the windward (leeward) side of the Sierra Nevada. This could be related to the mechanism identified by Siler and Roe (2014), although given the importance of mountain-top convection to summertime precipitation here, it may also result from changes in other mechanisms. In general, because the lee side of WUS mountain ranges are typically arid, these large and robust fractional increases in lee-side precipitation will likely have a significant impact on local water resources and ecology.

### 4.2 Changes in Extremes

The future fractional change in extreme (rx1day) precipitation is much more consistent across the WUS than for the mean, with intensified extremes occurring over most of the domain in both the parent and dynamically downscaled GCMs (Figure 9a,c, Fig S10 for individual GCMs). These changes in rx1day vary from roughly 0-12 %/K across the domain in the downscaled ensemble mean. In both the

GCM and downscaling cases, the spatial variations in the changes in rx1day can be traced in part to vertical velocity changes: Spatial correlations of -0.7 (-0.3) are found between rx1day precipitation and vertical-velocity changes in GCM (downscaling) experiments. However, the patterns of vertical velocity change are very different in the two cases. In the downscaling experiments, the largest intensification of rx1day occurs via vertical-velocity increases on the lee side of latitudinally-oriented mountain chains that are not resolved in the parent GCMs (Figure 8b,d). Over large parts of these areas, the increases are super-Clausius-Clapeyron ( >7%/K). This indicates that extreme precipitation intensifies at a greater rate than saturation specific humidity, commonly termed the thermodynamic component of extreme-precipitation scaling. Thus, WRF simulates greater dynamical intensification of extreme precipitation (e.g., Norris et al., 2019) than the GCMs, and in a distinct topographically-modulated pattern.

Next, we examine future changes in extreme heat, defined by the number of days exceeding the $99^{th}$ percentile of the historical daily-maximum surface air temperature (Tmax99). Consistent with extreme precipitation, these changes are normalized by global warming to account for the large intermodel spread in climate sensitivity (Figure 10, left). Averaged across the WUS, Tmax99 exceedances increase by $11.9 \pm 2.1$ days per year $K^{-1}$. California, Oregon, and Washington see increases of 4-7 days per year $K^{-1}$, with coastal areas see increases of less than 5 days per year $K^{-1}$. The power of dynamical downscaling is particularly evident, as the GCMs (top row, Fig. 10) cannot simulate (i) the correct coastline geometry, leading to an unphysical intrusion of maximized ocean-influenced Tmax99 exceedances, or (ii) the complex terrain of the WUS, which strongly modulates the snow coverage and subsequently the land surface sensitivity to warming. Additional examination reveals that the GCMs with the greatest regional mean warming are not necessarily the GCMs with the largest increase in

exceedances (Figs. S5, S12). This discrepancy may be due to GCM differences in simulation of synoptic-scale events that produce heat waves. Anthropogenic changes in such events may occur independently of mean temperature shifts (Fig. 7).

We find that the spatial pattern of Tmax99 exceedances is different from the pattern that arises assuming a shift to the temperature distribution equally to the mean warming (Fig. 9; middle). A mean shift significantly underpredicts the increase in exceedances by 3-4 days per year $K^{-1}$ across portions of California, Oregon, and Washington. Still greater underpredictions of future exceedances assuming a mean shift are seen across western Montana, Idaho, and portions of western Wyoming, particularly at higher elevations. Further south however, exceedances in Tmax99 are generally consistent with shifts in the mean. Exceedances are slightly overpredicted across portions of New Mexico and western Texas. This analysis highlights that intensification of extreme temperature will be affected by both a mean shifts and alterations of the tail of the temperature distribution. This is true in both GCMs and downscaled solutions, and the patterns in the right panels of Figure 10 are broadly similar. However, there is also significant spatial structure in the downscaling patterns not seen in the GCMs, indicating that local atmospheric dynamics and local land-atmosphere feedbacks play a role in shaping change in the right tail of the temperature distribution.

**Summary and Conclusions**

Future regional climate change remains difficult to project, given the low resolution of GCMs, particularly over a region of complex terrain such as the western U.S. In this study, we present a dataset containing sixteen CMIP6 models dynamically downscaled with WRF over the region from 1980 to

2100 at 9-km grid spacing: the Western U.S. Dynamically Downscaled Dataset (WUS-D3). The future projections are primarily based on the SSP3-7.0 high-emissions scenario, but we include two additional downscaled experiments with CESM2 of the SSP2-4.5 and SSP5-8.5 scenarios. An extensive evaluation of CMIP6 models' historical simulations over the western U.S. has been conducted (Krantz et al., 2021; Goldenson et al., 2023, in revisions) to identify the most suitable candidates for downscaling over this region. However, GCM selection was also based on the availability of data required to provide initial and boundary conditions to WRF. The optimal configuration of WRF over the western U.S. was established via an extensive evaluation of an ERA5-driven WRF run (Rahimi et al., 2022). Numerous other challenges of using the CMIP6 data to force WRF are outlined in the methods.

Aside from the obvious improved representation of spatial patterns of meteorological variables, there are many notable improvements of the downscaling over raw GCMs when compared to observations over the historical period. For example, the WRF simulations largely correct for major wet biases (~100%) in the raw GCMs over Nevada, Wyoming, Montana, and Colorado. These bias reductions apply to both winter and summer, depending on the state. Moreover, the GCMs completely fail to represent the summertime precipitation maximum over Arizona and New Mexico, which is corrected in the downscaled experiments, albeit with some large wet biases therein. The WRF simulations also correct large summertime warm biases over much of the domain, particularly the interior states, due to the improved representation of terrain and resulting snowpack improvements. Finally, extreme precipitation (measured by rx1day) is greatly increased (generally about doubled) from the GCM values. In some cases, this amounts to wet biases from WRF, according to PRISM, but these

apparent biases are mostly at high elevations where observational uncertainties are maximized (Lundquist et al., 2019).

There are, however, biases that remain from the parent GCMs and in some cases are exacerbated. For example, the GCMs generally overestimate winter precipitation along the west coast, which likely results from unrealistically high moisture contained within atmospheric rivers (Norris et al., 2021) and other GCM biases transmitted to WRF. And in the WRF simulations, these wet biases are amplified, likely as excessive moisture is forced up steeper orographic gradients than in the GCMs. Also, unlike the GCMs, the downscaled experiments are generally too cold compared to PRISM, particularly in winter, with some states exhibiting as much as 5 K bias. These results are comparable to Rastogi et al. (2022), who used a different regional climate model, implying that inherited GCM biases may be to blame.

The future downscaled climate change signals are shaped in physically credible ways by the regional model's more realistic coastlines and topography. Large-scale warming patterns are generally preserved from the parent GCMs, but with enhanced warming adjacent to high terrain during winter and spring and over high elevations during summer. This locally enhanced warming occurs where relative snow losses are maximized in the future, a feature that cannot be captured at the GCMs' coarse resolution. Meanwhile, precipitation patterns undergo much greater transformation with downscaling. Although WRF preserves the broad pattern of subtropical drying and midlatitude wetting, WRF simulates additional local precipitation changes. In particular, mean precipitation changes are often consistent with wetting on the windward side of mountain complexes, as warmer, moister air masses are uplifted orographically during precipitation events, similar to Huang et al. (2020). There are large

fractional precipitation increases on the lee side of mountain complexes, consistent with the theoretical work of Siler and Roe (2014). This could lead to significant changes in water resources and ecology across these arid landscapes. The intensification of precipitation and temperature extremes is also modified in significant ways by dynamical downscaling. Over complex terrain, precipitation extremes scale at much greater rates, on the order of 12%/K. This greater scaling in WRF is likely due to greater dynamical enhancement of extreme precipitation over mountain ranges, as evidenced by the intensification of vertical velocity increases conditioned on extremes. Temperature extremes also intensify, as measured by future exceedances of historical 99th-percentile surface air temperature, per degree global warming. These are on the order of +5 days year$^{-1}$ K$^{-1}$ along the west coast and approaching 15 days year$^{-1}$ K$^{-1}$ in the interior west. The simulated changes are mostly greater than that predicted by a simple mean shift of the temperature distribution, indicating the effect of an extension of the right tail. The imprint of topography is evident in this change in the temperature distribution's shape, indicating the importance of local atmospheric dynamics and land surface feedbacks.

WUS-D3's constitutes the first comprehensive dataset of landscape-resolving climate projections over the western U.S. Although only temperature and precipitation projections have been evaluated here, the dataset includes all 2-D and 3-D meteorological and land-surface variables at 6-hourly resolution with a auxiliary datastream of more than 20 land-surface variables needed to drive downstream models (e.g., hydrology) offline. Thus, it represents a unique opportunity to explore potential future changes to a wide diversity of weather/climate phenomena over the region. These include but are not limited to atmospheric rivers, the North American monsoon, summer convective storms, intense heat waves, wildfire-related downslope winds, and ventilation by sea breezes. Moreover,

Geoscientific Model Development Discussions Open Access EGU

these data may be used to drive offline and calibrated hydrology and fire-weather models to obtain more detailed projections of water resources/flooding and wildfire. Nevertheless, there are biases in the downscaled simulations, briefly documented here, which should be understood and appreciated when using the data for future projections. We strongly encourage the community to use these results with other dynamically and statistically downscaled products to develop risk assessments and bound uncertainty. Such intercomparisons of different downscaled products are critical to assessing a product's usefulness and applicability.

**Data Availability**

All downscaled data for WUS-D3, including the full 6-hourly WRF datastream (Tier 1), hourly data for select land-surface variables (Tier 2), and a daily post-processed datastream (Tier 3) are located in the following open-data bucket on Amazon S3: s3://wrf-cmip6-noversioning/ at https://registry.opendata.aws/wrf-cmip6/. These data are completely open and free to the public. We have also developed a technical access and usage document that details these three data tiers which can be found at https://dept.atmos.ucla.edu/sites/default/files/alexhall/files/aws_tiers_dirstructure_nov22.pdf and on ResearchGate at (https://www.researchgate.net/publication/374504614_Data_tier_descriptions_directory_structure_and_data_access_of_the_Western_US_Dynamically_Downscaled_Dataset_WUS-D3_version_1; DOI:10.13140/RG.2.2.11385.85609). As recommended in the document, these data are most easily downloaded when using Amazon Web Service's (AWS') Command Line Interface (CLI) or with wget. An example is presented in the technical access and usage document.

**Code Availability**

Individualized codes were developed to create the intermediate binary files for each GCM before ingestion into WRF. As such, we have created uploaded these files into the Amazon S3 bucket in a subdirectory labeled 'downscaling_codes'.

**Team List**

Stefan Rahimi, Lei Huang, Alex Hall, Naomi Goldenson, Will Krantz, Benjamin Bass, Chad Thackeray, Henry Lin, Di Chen, Eli Dennis, Emily Slinskey, Sara Graves, Surabhi Biyani, Stephen Copper, and Elease Liu Stemp from the UCLA Center for Climate Science contributed to the evaluation of this product, along with Zachary Lebo from the University of Oklahoma nad Ethan Collins from the University of Wyoming.

**Author Contributions**

SR and LH executed the simulations; SR and AH wrote the manuscript draft; JN, AH, and ZL edited the manuscript, NG, WK, and LH developed workflow to evaluate GCM performance and download GCM outputs, while BB, CT, HL, DC, ED, EC, ZL, and ES led analyses to explore the downscaled results in more detail.

**Competing Interests**

The authors declare that they have no conflict of interest.
**Funding and Acknowledgements**

We acknowledge the supporting funding agencies supporting this project: the Department of Energy's HyperFACETSS (A23-1053-S003) project, the Strategic Environmental Research and Development Program (SERDP) under Project , the California Energy Commission (EPC-20-006), and the University of California's Climate Ecosystems Future (LRF-18-542511) project. We also thank the computational support through the NCAR-Wyoming Supercomputing Center (NWSC) and the Computational and Information Systems Laboratory (CISL). We also acknowledge the TORNERDO consortium, as well as Cora for writing assistance.

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

## Tables and Figures

**Table 1. List of GCMs dynamically downscaled in this study. Approximate near-equatorial latitude-longitude resolutions are given.**

| GCM | Variant | Center | Resolution | Source | SSP | SST mod? |
|---|---|---|---|---|---|---|
| ACCESS-CM2 | r5i1p1f1 | Commonwealth Scientific and Industrial Research Organization | 1.25°x1.25° | (Bi et al., 2020) | 3-7.0 | yes |
| CanESM5 | r1i1p2f1 | Canadian Climate Center | 2.8°x2.8° | Swart et al., (2019) | 3-7.0 | yes |
| CESM2* | r11i1p1f1 | National Center for Atmospheric Research | 0.94°x1.25° | Danabasoglu et al., (2020) | 2-4.5, 3-7.0, 5-8.5 | no |
| CNRM-ESM2-1 | r1i1p1f2 | Centre Europeen de Recherche et de Formation Avancee en Calcul Scientifique | 1.4°x1.4° | Séférian et al., (2019) | 3-7.0 | no |
| EC-Earth3 | r1i1p1f1 | EC-Earth Consortium | 0.7°x0.7° | Döscher et al., (2022) | 3-7.0 | yes |
| EC-Earth3-Veg | r1i1p1f1 | EC-Earth Consortium | 0.7°x0.7° | Döscher et al., (2022) | 3-7.0 | yes |

| GCM | Variant | Center | Resolution | Source | SSP | SST mod? |
|---|---|---|---|---|---|---|
| FGOALS-g3 | r1i1p1f1 | Chinese Academy of Sciences | 2°x2° | Li et al., (2020) | 3-7.0 | yes |
| GISS-E2-1-G | r1i1p1f2 | National Aeronautic and Space Administration | 2°x2.5° | Kelley et al. (2020) | 3-7.0 | yes |
| MIROC6 | r1i1p1f1 | Japan Agency for Marine-Earth Science and Technology | 1.4°x1.4° | Tatebe et al. (2019) | 3-7.0 | yes |
| MPI-ESM1-2-HR | r7i1p1f1 | Max Planck Institute | 0.94°x0.94° | Gutjahr et al. (2019) | 3-7.0 | yes |
| MPI-ESM1-2-LR | r7i1p1f1 | Max Planck Institute | 1.9°x1.9° | Mauritsen et al., (2019) | 3-7.0 | no |
| NorESM2-MM | r1i1p1f1 | NorESM Climate modeling Consortium | 0.94°x1.25° | Seland et al. (2020) | 3-7.0 | yes |
| TaiESM1 | r1i1p1f1 | Research Center for Environmental Changes | 0.94°x1.25° | Wang et al. (2021) | 3-7.0 | yes |

| GCM | Variant | Center | Resolution | Source | SSP | SST mod? |
|---|---|---|---|---|---|---|
| UKESM1-0-LL | r2i1p1f2 | Met Office Hadley Centre | 1.25°x1.25° | Sellar et al., (2020) | 3-7.0 | yes |

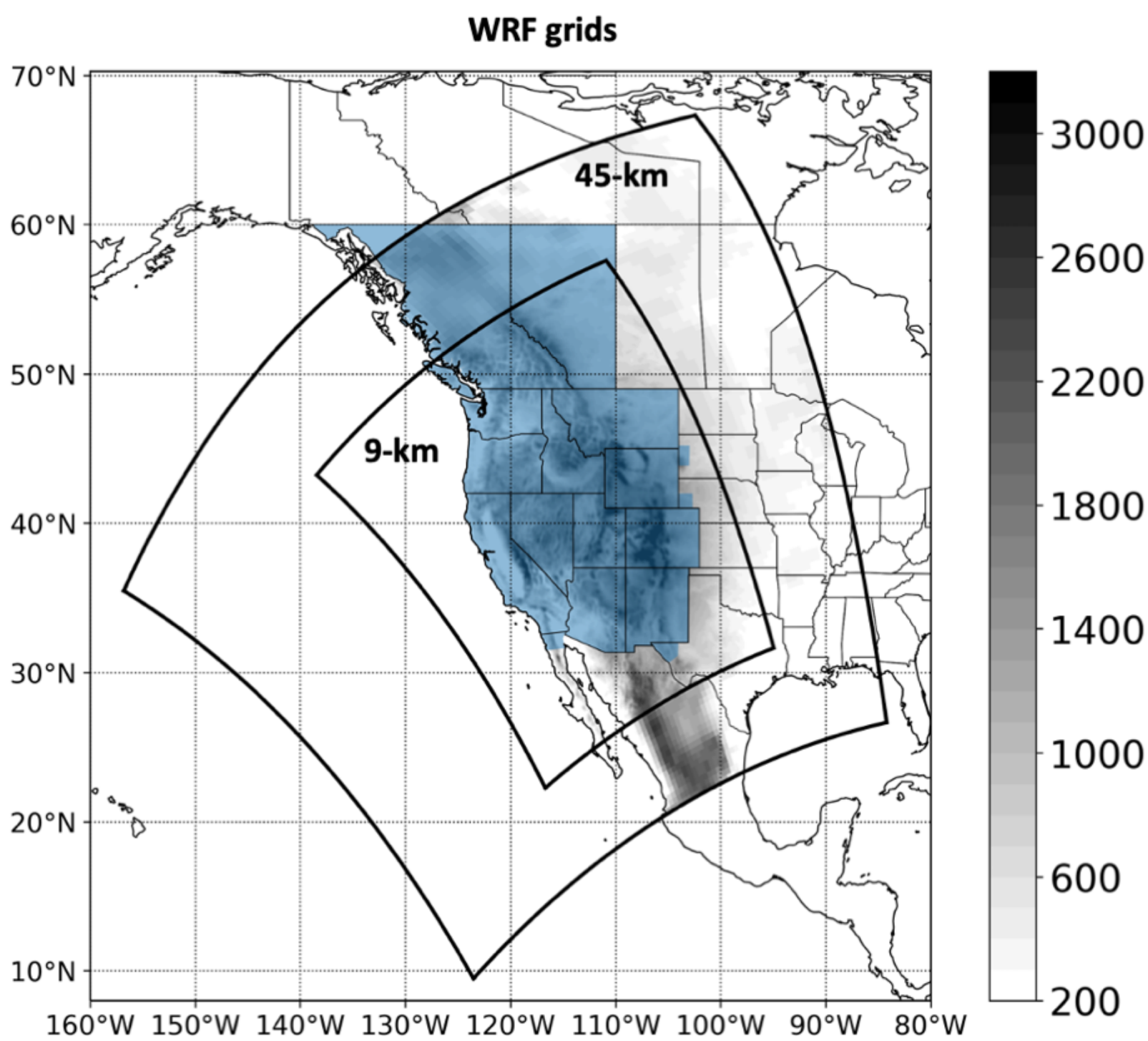

**Figure 1: WRF grids used in this study. Topography [m] is shaded to its highest resolution, and the blue shading indicates the Western Electricity Coordinating Council (WECC) coverage area.**

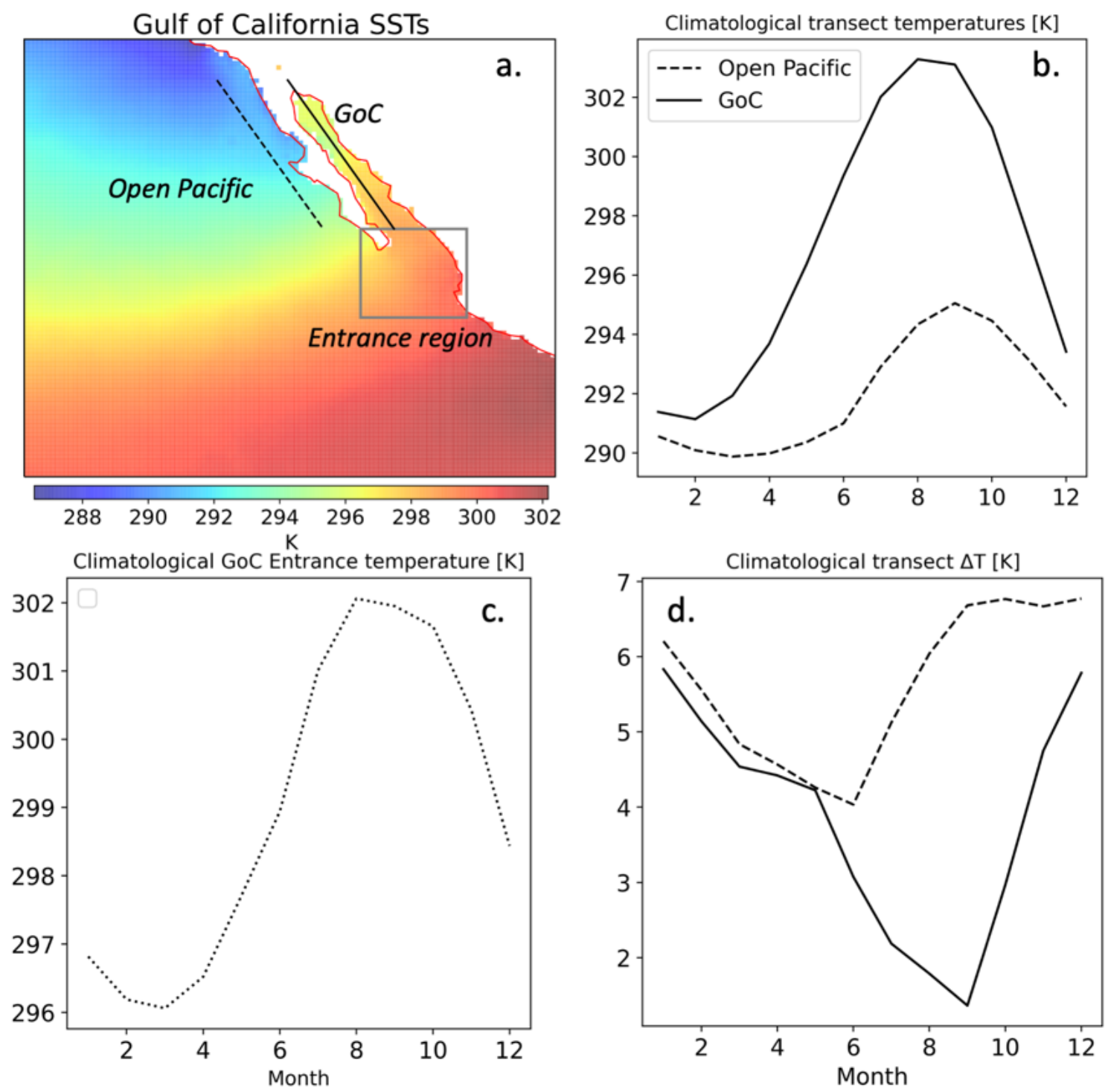

**Figure 2. Panel (a) Climatological (1980-2014) mean SSTs from ERA5 along with transects across the Gulf of California (GoC; solid black curve) and Open Pacific (dashed black curve). The gray bounded zone is our GoC entrance region. Panel (b) shows the latitudinally weighted transect-mean temperatures from the Open Pacific and GoC. Panel (c) shows the area-weighted GoC entrance region temperature, while panel (d) depicts the temperature gradient along the Open Pacific and GoC transects from northwest to southeast.**

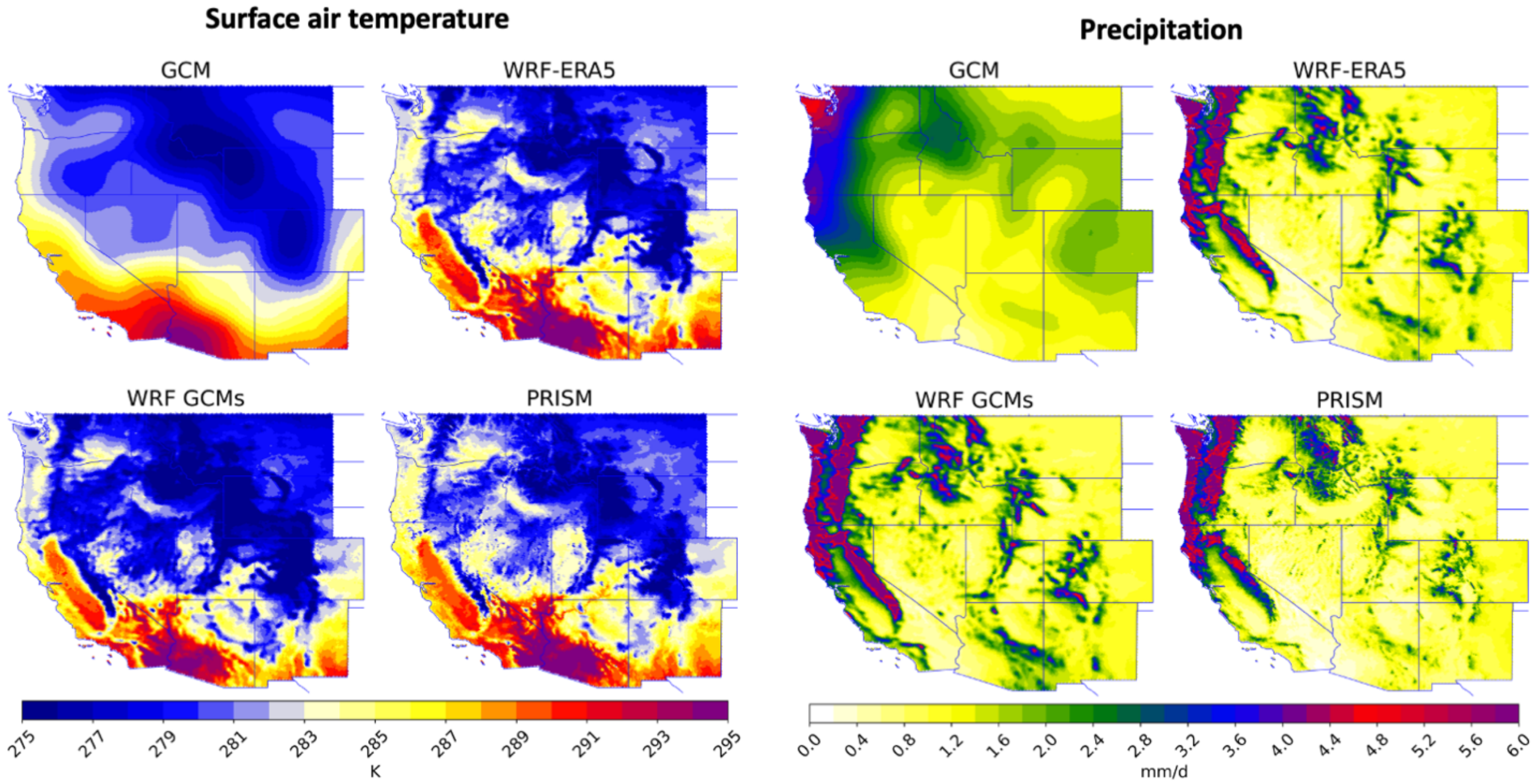

**Figure 3. 1981-2010 annual mean (left) surface air temperature [K] and (right) precipitation rate [mm $d^{-1}$] from the native GCMs (GCM; 14-GCM ensemble mean), dynamically downscaled ERA5 (WRF-ERA5), dynamically downscaled GCMs (WRF GCMs; 14-member ensemble mean), and PRISM. All GCM and PRISM data are interpolated from their native grids to the 9-km WRF grid.**

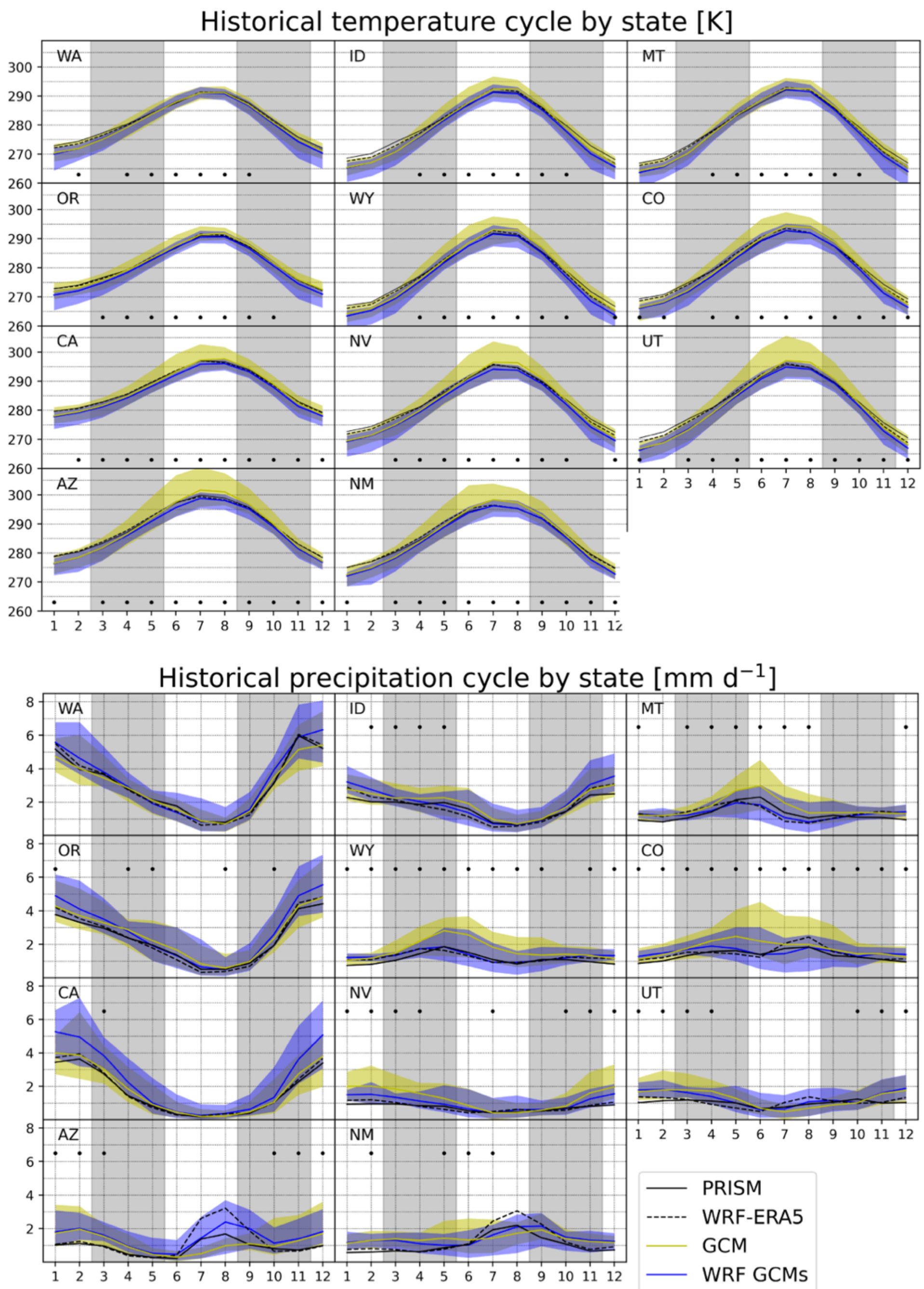

835 **Figure 4. 1981-2010 seasonal cycles of state-mean surface air temperature [K] and precipitation [mm d$^{-1}$] from native GCMs (GCM), dynamically downscaled ERA5 (WRF-ERA5), dynamically**

**downscaled GCMs (WRF GCMs; 14-member ensemble mean), and PRISM. The parent and downscaled GCM ensemble spreads are presented in yellow and blue shading, respectively. Black circles indicate months where the dynamically downscaled spread is smaller than the parental GCM spread.**

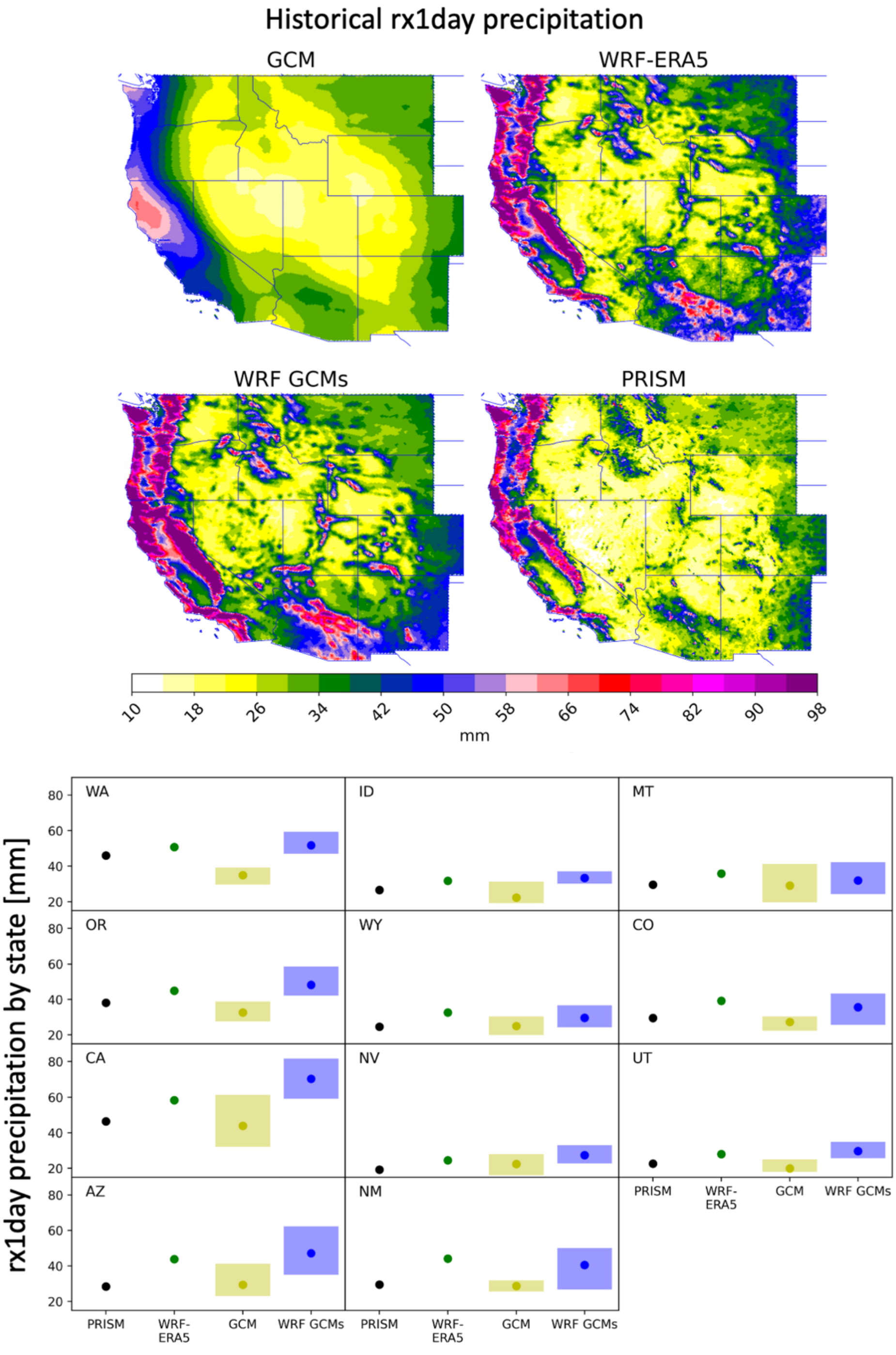

**Figure 5. Historical (1981-2014) mean rx1day (annual-maximum daily precipitation) precipitation amounts [mm] from native GCMs (14-GCM ensemble mean), dynamically downscaled ERA5 (WRF-ERA5), dynamically downscaled GCMs (WRF GCMs; 14-member ensemble mean), and PRISM. The top figure presents the spatial distribution of rx1day precipitation, while the bottom figure presents rx1day precipitation amounts averaged across each western U.S. state. Ensemble mean values are presented as colored circles, while the GCM spread in rx1day values is shaded. GCM data were interpolated to a 1° rectilinear grid before computations.**

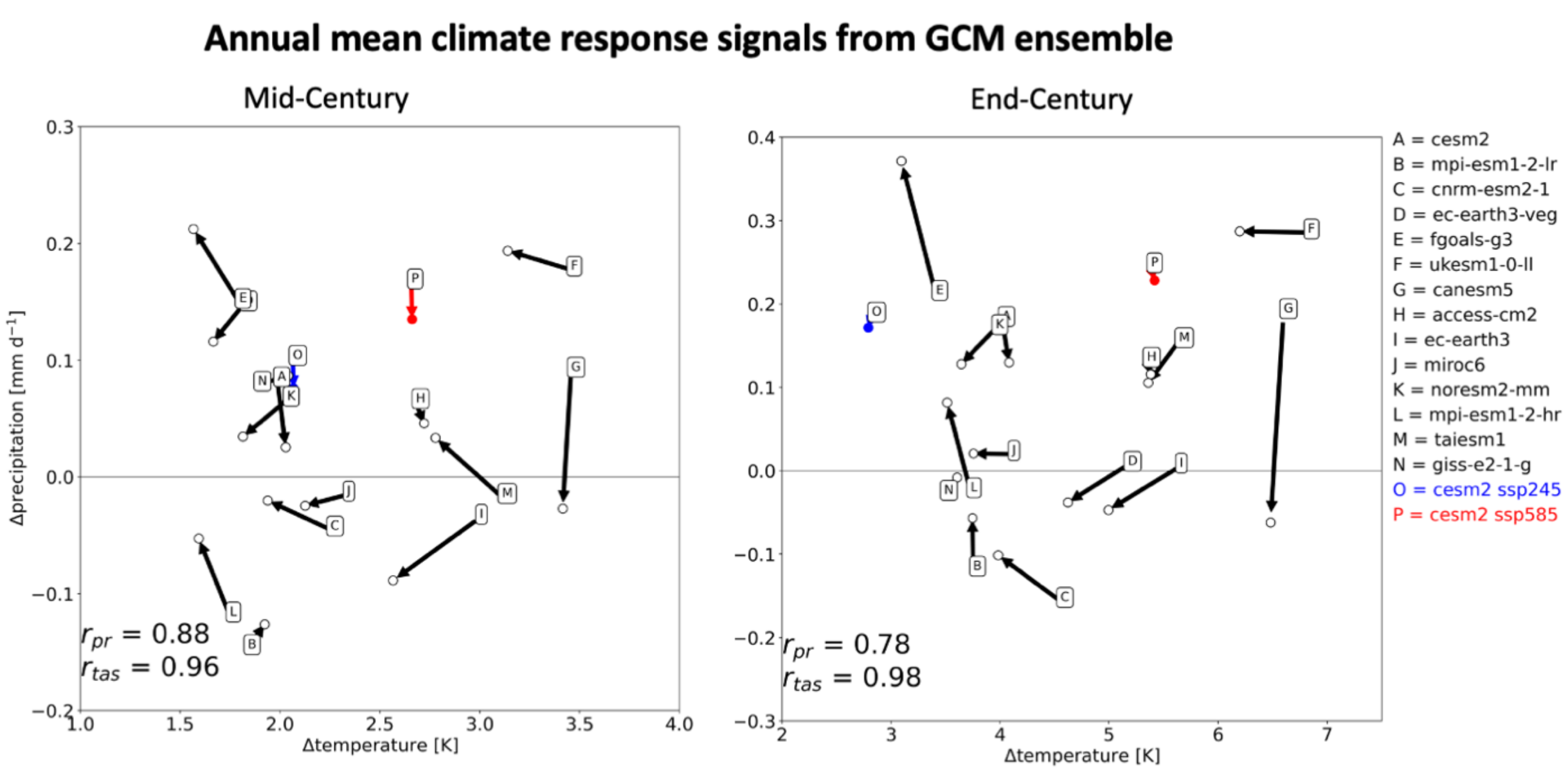

**Figure 6. Future climate response for parent GCMs (indicated by lettering) and their downscaled counterparts (indicated by open circles) on the 9-km WRF grid averaged across 11 western U.S. states. Non-colored circles are for SSP3-7.0 projections only, while blue (red) circles represent the**

855 **SSP2-4.5 (SSP5-8.5) projections. Arrows point away from parent GCMs towards downscaled counterparts.**

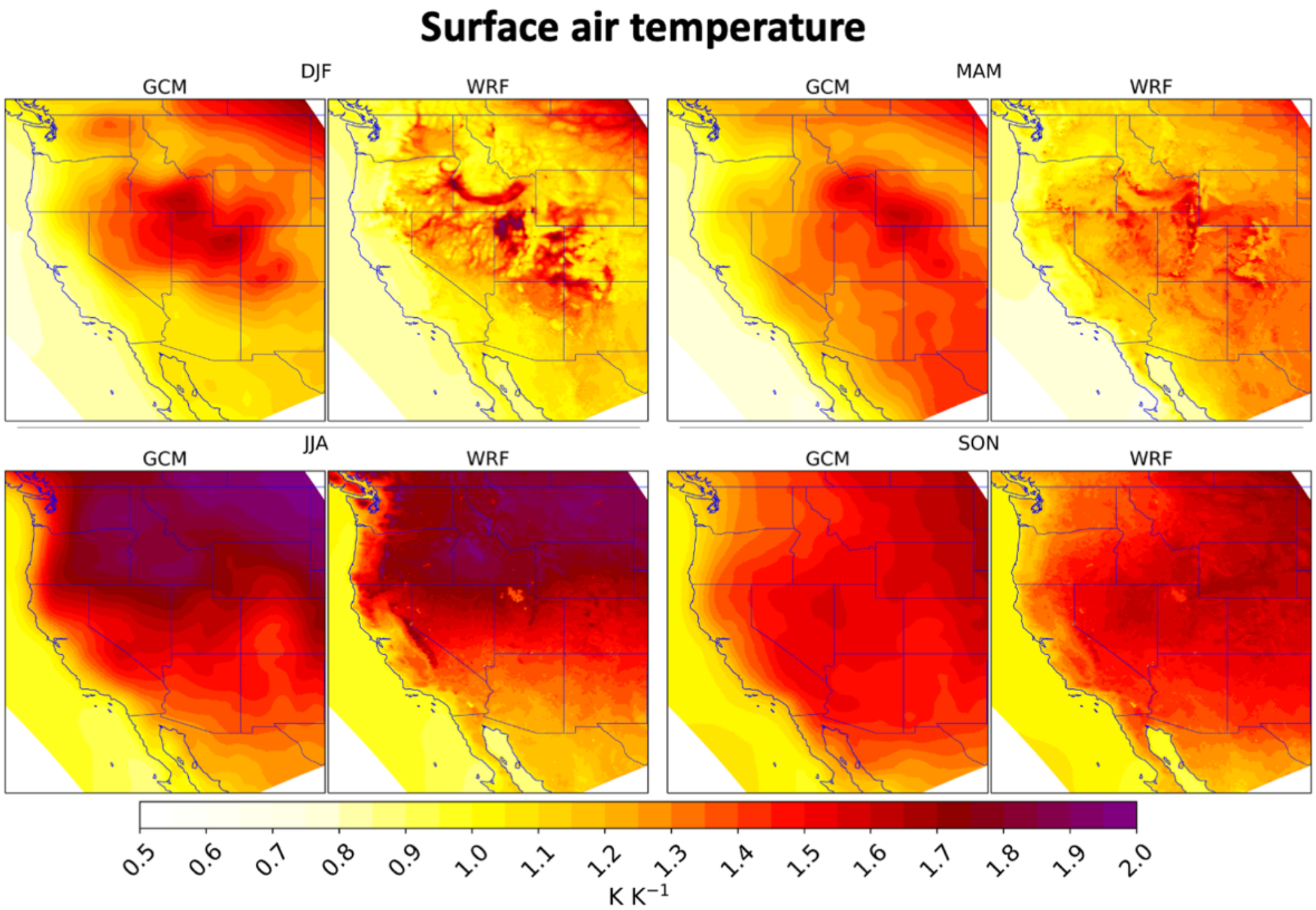

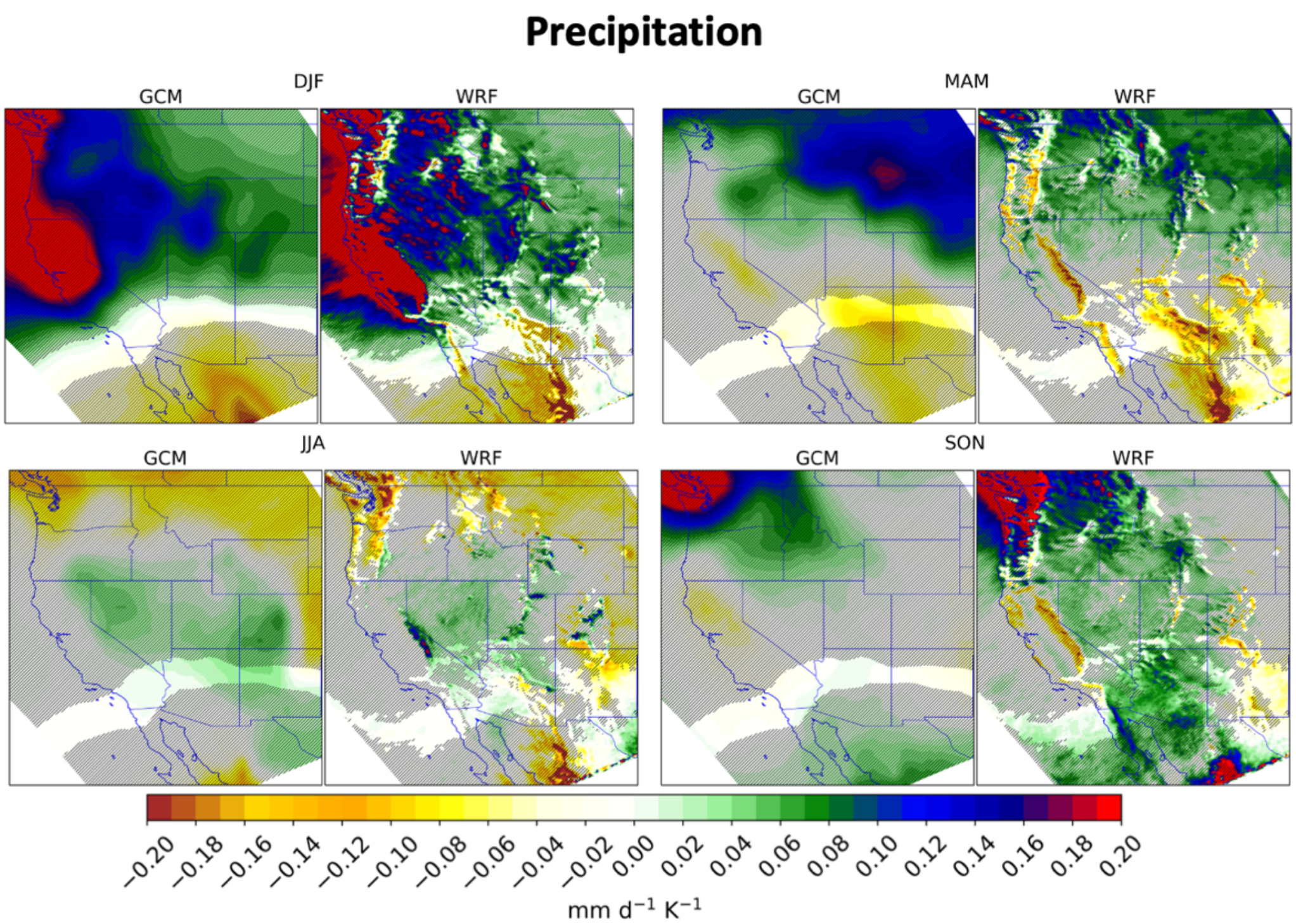

**Figure 7. Ensemble-mean future changes in (top) seasonal surface air temperature [K $K^{-1}$] and (bottom) precipitation per degree of global warming [mm $d^{-1}$ $K^{-1}$] from 16 downscaled GCMs. Hatching indicates statistical significance to the 95% confidence interval when grid point distributions are subjected to a two-sided Student's t-test. Stippling is not included for temperature because every grid point returns a p value smaller than 0.05.**

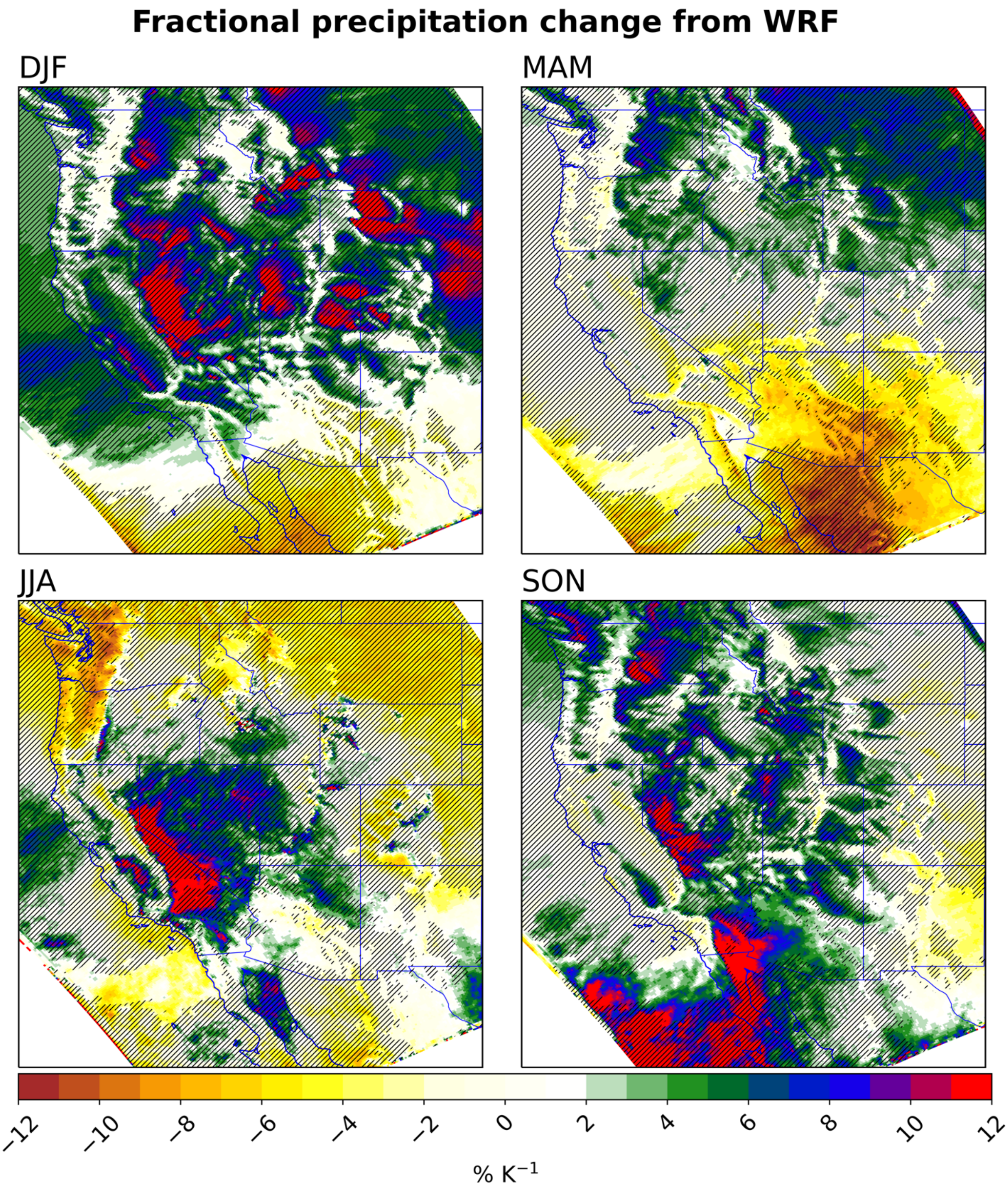

**Figure 8. 16-GCM-mean future changes in dynamically downscaled fractional precipitation normalized by the amount of global warming [% $K^{-1}$]. Hatching indicates statistical significance to the 95% confidence interval when grid point distributions are subjected to a two-sided Student's t-test. Stippling is not included for temperature because every grid point returns a p value smaller than 0.05.**

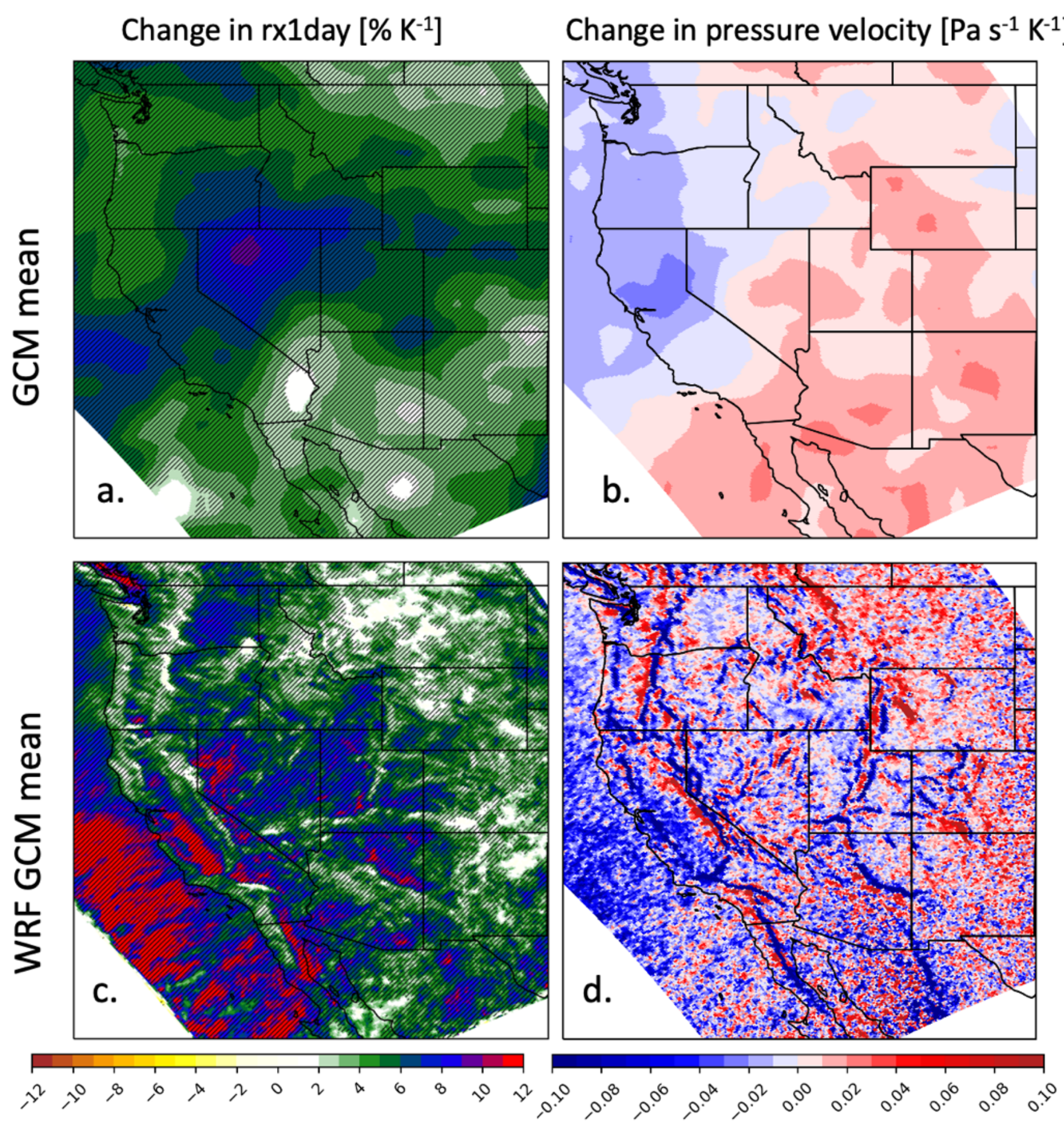

**Figure 9. Future response in (left) rx1day precipitation [% $K^{-1}$] and 500 hPa pressure velocity [Pa $s^{-1}$ $K^{-1}$] conditioned on rx1day occurrence for the (top) GCM and (bottom) WRF ensembles per degree of global warming [% $K^{-1}$]. Hatching in left panels denotes areas with p values less than 0.05. Due to the lack of GCM data with daily vertical velocity outputs, we use a 14-GCM (9-GCM) mean for rx1day (pressure velocity); the WRF patterns of rx1day generally are similar for a 9-GCM mean (Figure S11).**

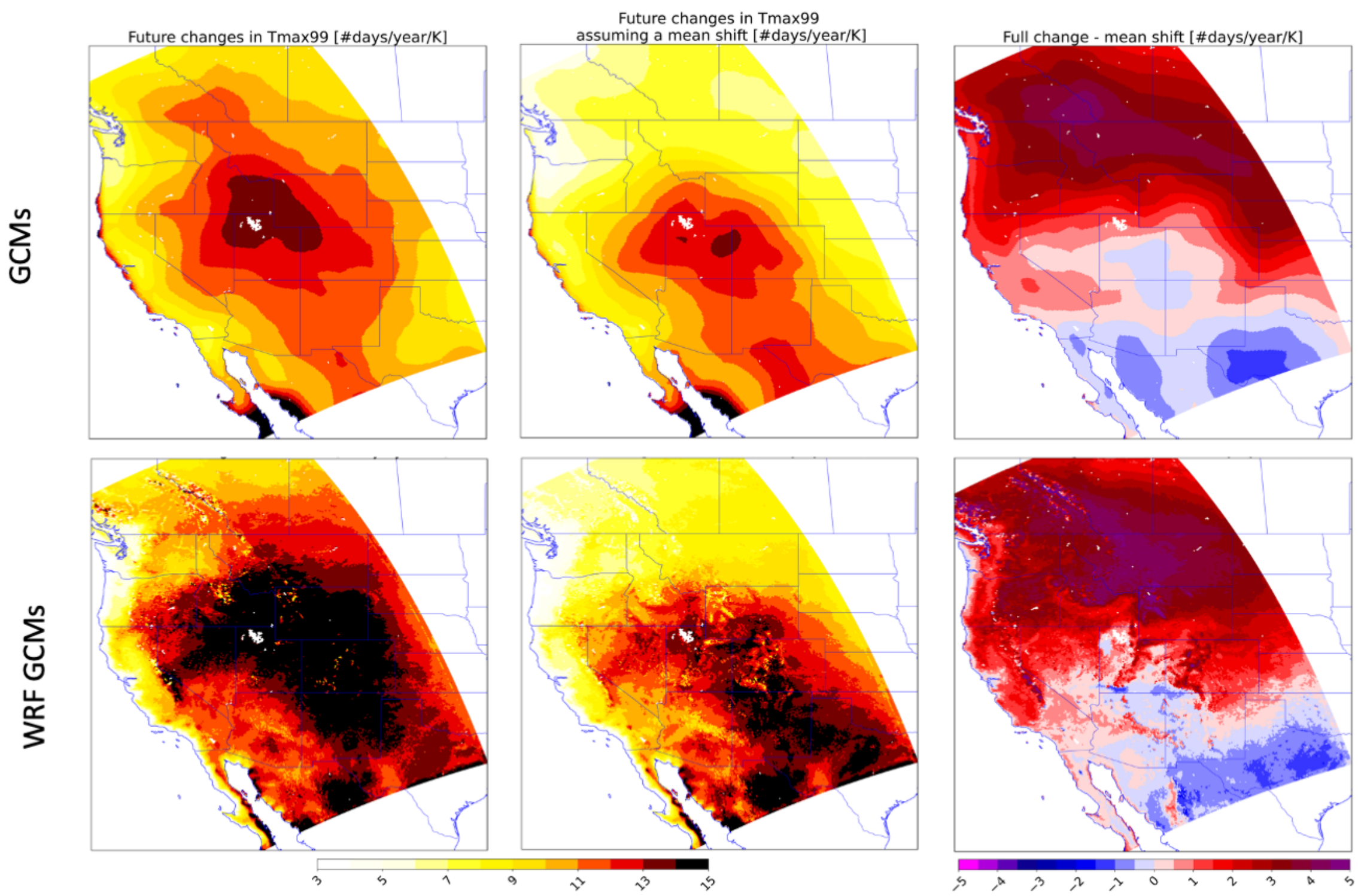

**Figure 10. Future changes in maximum daily 99$^{th}$ percentile surface air temperature (Tmax99) per degree of global warming [days year$^{-1}$ K$^{-1}$] considering the full change (left) and assuming a mean shift in the temperature distribution (middle). The right panel presents the difference between the left and center panels. Parent GCM (WRF GCM) calculations utilize a 11-member (16-member) ensemble. When using the same 11-member ensemble, the WRF panels look similar (Figure S13).**