# Peer review of "An Overview of the Western United States Dynamically Downscaled Dataset (WUS-D3)"

_Geoscientific Model Development, 2023_

## Referee Comment (RC1)

Comments for the manuscript "*An Overview of the Western United States Dynamically Downscaled Dataset (WUS-D3)*"

**General comments**

The authors provided an overview of the WUS-D3 dataset for the complex terrain western United States using the dynamical downscaling method based on 14 CMIP6 GCMs historical and SSP3-7.0 scenario simulations plus two projections SSP2-4.5 and SSP5-8.5 of one GCM (CESM2). The atmospheric model WRF with the horizontal resolution of 9 km and 39 vertical levels was used to conduct downscaling simulations. The authors described the challenges of producing WUS-D3 dataset, including GCM selection and technical issues, as well as an evaluation for the simulations' realism by comparing historical results to temperature and precipitation observations. They concluded that because of its high resolution, comprehensiveness, and representation of relevant physical processes, this dataset presents a unique opportunity to evaluate societally relevant future changes in western U.S. climate.

The method is unusual: "We downscale each GCM year separately and in parallel; at the beginning of each downscaling period (on August 1), the RCM is initialized to the driving GCM state."

The reason given is not yet convincing: "WRF's parallelization procedure, which is advantageous for executing simulations in weeks instead of years, is performed to the detriment of time continuity in simulating the surface and subsurface runoff with high precision." With that deficit, the soil moisture memory is neglected in these simulations. Not using a continuous simulation, will lead to unrealistic jumps in storage variables, especially in soil moisture or the snowpack. This, in turn, may cause effects in other variables, such as evapotranspiration, latent heat flux, albedo and 2m temperature. Hence, the authors should discuss the potential deficits of the dataset implied by not using a transient simulation.

The analyses do not state much about the difference between WRF-GCMs with SSP2-4.5, 3-7.0 and 5-8.5. How to explain in Mid-Century (Figure 6, left) that the mean temperature change of SSP2-4.5 of CESM2 (O) is greater than SST3-7.0 (A)? And why in End-Century (Figure 6, right), the SSP3-7.0 (A) has the longer arrow than O and P?

Any outlook for bias reduction/correction? 5 K cold bias seems to be quite large. Implications of this large bias should be discussed.

Discussion and outlook: as the considered area (WUS) is next to the open Pacific, has any atmosphere-ocean coupled model been applied for this region for downscaling CMIP6? What would be the role of the Pacific Ocean on regional climate over WUS, at least in the 45 km set up which cover half of ocean?

I suggest accepting the paper for publication after minor revisions are made.

**Minor Comments**
- Lines 66-76: Arguments are not clear. Please rephrase the paragraph.
- Line 92: Please list here names of the 11 states and display them on Fig.1. Not all readers are familiar with their locations.
- Line 106-107: "SSP-2-4.5 and SSP-5-8.5" should be "SSP2-4.5 and SSP5-8.5" to be consistent with SSP3-7.0 and themselves on other pages.

- Line 126: Sentence is not clear: "To address this issue, we propose that the atmospheric fields from WRF be used to drive offline and calibrated hydrology models that are continuous". Please rewrite it.
- Line 128: Which kind of aerosols were used in WRF for these simulations? It's expected to use transient aerosols for such historical-scenario simulations.
- Line 133: Why not using the transient land-use/land-cover from CMIP6?
- Line 190: "if at all in CMIP6 GCMs": not clear what the authors mean here
- Line 262: What is the resolution of PRISM dataset?
- Line 272: Should the "black circles" be plotted in red to increase eyes-catching effect?
- Line 272-273: Any explanation/speculation for the result of "Exceptions are noted across some western states, especially in winter"?
- Line 277-278: What does "meaningful subregional biases of hundreds of percent" mean here?
- Line 290-291: Sentence is not clear
- Line 292: How was "rx1day" defined/determined?
- Line 324-325: "Despite a positive mean change, a handful of simulations suggest drying across the region.": where does the information come from?
- Line 326: "For warming amounts,…"? Should it be "For temperature change,…" as on the line 328, the "precipitation change" is mentioned.
- Line 341: Typo on title of section 4.1 "Patters" instead of "Patterns"
- Line 385: What are "ensemble-mean fractional precipitation changes"?
- Line 433-434: Sentence is unclear
- Line 434: Should it be (Fig.10, middle) instead of (Fig.9, middle)?
- Figure 3 (and some elsewhere): It would be easier to compare if the PRISM dataset figure and the WRF-ERA5 figure locations are exchanged
- Figure 8 and Figure S9 caption: Should "Stippling is not included for temperature because every grid point returns a p value smaller than 0.05." be removed?
- Figure 8 (line 414): There is no a, b, c, d on the Figure 8
- Figure S2caption: Should move the unit [K] after the word "biases" like this: "11-state-mean biases [K] are presented beneath each GCM label."
- Figure S5 caption: Should add the unit [K] after "annual-mean surface air temperature"
- Figure S7 caption: Typo in "the" 16-GCM mean. Should remove "1 April" as the figure shows three months (Jan Apr, Jul)
- Figure S10 caption: Should move [mm d$^{-1}$] after "rx1day precipitation"
- Figure S11 caption: [% K$^{-1}$] should be located after "rx1day precipitation"

---

## Referee Comment (RC2)

**Review of the manuscript:**

**"An Overview of the Western United States Dynamically Downscaled Dataset (WUS-D3)"**

by *Stefan Rahimi, Lei Huang, Jesse Norris, Alex Hall, Naomi Goldenson, Will Krantz, Benjamin Bass, Chad Thackeray, Henry Lin, Di Chen, Eli Dennis, Ethan Collins, Zachary J. Lebo, Emily Slinskey, and the UCLA Center for Climate Science Team*

In this manuscript, the authors investigated the performance of dynamically downscaled simulations of climate conditions over the western U.S. from 1980 to 2100. The simulations were conducted using the WRF model driven by sixteen selected CMIP6 GCMs for various SSP scenarios. The WRF model was run on two nested domains with resolutions of 45 km and 9 km, yearly initiated with a one-month spin-up for the land surface, spanning from August 1 to September 1 of the next year. The authors have addressed many aspects related to the dynamical downscaling technique applied in climate change projection. The manuscript is very interesting, and it's worth publishing in the Geoscientific Model Development. However, I would like to request the authors clarify the following points.

1) Since the WRF model was initiated yearly, soil moisture data for the land model were provided by GCMs. These soil moisture data may be significant differences compared to those in case continuous running of the model. Have the authors conducted tests to assess the impact of this difference on the model outputs?

2) Why did the authors used one-way nesting from the parent domain (45-km) to the inner domain (9-km) instead of employing two-way nesting to gather feedback on local features that could benefit from a finer resolution?

3) I'm uncertain about the method the authors used to determine the projected changes in temperature ($K\ K^{-1}$) and precipitation ($mm\ d^{-1}\ K^{-1}$) per degree of global warming in Figure 7. Just to clarify, are these changes being calculated only for the end of the 21st century? Please let me know if I understood correctly. If so, is the calculation of "global warming" based on the ensemble mean derived from all sixteen GCMs for the entire globe?

---

## Community Comment (CC1)

**Manuscript:** An Overview of the Western United States Dynamically Downscaled Dataset (WUS-D3)

The paper presents a comprehensive study introducing downscaling work to a 9 km resolution for 16 CMIP6 GCM experiments using the WRF model. The manuscript well describes the methodology and the WUS-D3 dataset. I recommend publishing the paper after some minor revisions as follows.

Comments:

1. GCM selection: the authors outlined 6 processes considered in the evaluation and selection of GCMs. While they refer to the ranking methodology in a technical note (Krantz et al. 2021) and a paper currently under revision (Goldenson et al. 2023, in revisions), I recommend providing more information on two key aspects. Firstly, elaborate on the process selection – explain why these 6 processes were chosen; why extreme precipitation across California is included among the selected processes, given the coarse resolution of GCMs and the fact that this diagnostic variable might not play a role in the ICBC of the downscaling framework. Secondly, provide more details on the ranking methodology: clarify how these 6 processes are considered in the final ranking; are they equally weighted? How do temporal and spatial patterns contribute to the selection process?

2. L330: The authors stated that "Interestingly, downscaling generally reduces warming (leftward pointing arrows)" and hypothetically attributed it to the reduced snow albedo feedback with downscaling. I recommend that the authors prove this hypothesis by comparing the snow outputs of both WRF and GCMs.

3. The authors conducted a more in-depth analysis of the changes in rx1day and tmax99. However, there is no explanation as to why only these two indices, among many possible extreme indices, were selected. Furthermore, why did the authors opt for the absolute index (rx1day) when analyzing rainfall, while choosing the percentile index for temperature.

Minor comments:

1. L210, 215 should refer to Table 1's last column. The caption of Table 1 should also provide an explanation of the last column (SST mode)
2. Please add the names of locations mentioned in the text to Figure 1, such as California's Central Valley, Sierra Nevada, and state names, …

---

## Author Comment (AC3)

**Responses to RC2:** GMD-2023-162
Stefan Rahimi et al.

The reviewer comments are presented followed by underlined author responses.

In this manuscript, the authors investigated the performance of dynamically downscaled simulations of climate conditions over the western U.S. from 1980 to 2100. The simulations were conducted using the WRF model driven by sixteen selected CMIP6 GCMs for various SSP scenarios. The WRF model was run on two nested domains with resolutions of 45 km and 9 km, yearly initiated with a one-month spin-up for the land surface, spanning from August 1 to September 1 of the next year. The authors have addressed many aspects related to the dynamical downscaling technique applied in climate change projection. The manuscript is very interesting, and it's worth publishing in the Geoscientific Model Development. However, I would like to request the authors clarify the following points.

1. Since the WRF model was initiated yearly, soil moisture data for the land model were provided by GCMs. These soil moisture data may be significant differences compared to those in case continuous running of the model. Have the authors conducted tests to assess the impact of this difference on the model outputs?

In short, yes – tests were conducted to assess the impact of different spin-up periods.

The assumption that one month of soil spin-up is sufficient for climate change simulations is a massive one and deserves scrutiny (see comment by RC1), and the assumption should be regarded as a substantial limitation of WUS-D3. From the outset, we committed to some type of parallelization strategy to reduce integration times similar to other studies (e.g., the previous works of Zobel et al. below). However, we were not initially sure how much spin-up time was to be used nor how to parallelize. We eventually justified a one-month spin-up in reanalysis-direven tests in Rahimi et al. (2022), Here, we conduycted two year-long test experiments for water year 2010. In case 1, we used a single month of spin-up, and in test 2, four years of spin-up were integrated. Broadly speaking, we found there to be minimal differences in simulated soil moisture, soil temperature, surface air temperature, and precipitation between the two cases.

We are wary of the spin-up issues and resulting discontinuities in land-surface variables. For example, snow in WUS-D3 simulations is generally far too wet over the historical period, a feature common to different GCMs. By the end of each simulated year (31 August), snow does not completely melt out at all locations, leading to a discontinuity in the snow fields between 31 August and 1 September. Across these areas, this results in discontinuities in surface energy fluxes as the reviewer suggests. We have thus added Sec. 2.6 to the manuscript cautioning end-users about this issue:

'        Despite one month of spin-up in parallelized yearly WRF experiments, our adopted spin-up strategy neglects high-resolution soil memory on time scales greater than one month. This assumption may be particularly problematic across regions where a transient simulation is necessary to equilibrate the soil conditions to a state which

properly resolves the local-scale land-atmospheric coupling. For instance, some grid points do not see complete meltout of snow by 31 August 1993, but since data is retained from 1 September 1993 onwards, there are instances where discontinuities in surface snow coverage exist. This leads to discontinuities in surface energy variables (e.g., sensible heating; not shown). We encourage end-users of WUS-D3 to be wary of this pitfall. To alleviate this discontinuity, we propose that the atmospheric temperature, precipitation, surface radiative fluxes, winds, and specific humidity from WRF be used to drive offline calibrated hydrology models that are time-continuous and can be integrated much more rapidly (e.g., Bass et al., 2023). We acknowledge that this approach is inadequate across regions with a strong land-atmosphere coupling.

2. Why did the authors used one-way nesting from the parent domain (45-km) to the inner domain (9-km) instead of employing two-way nesting to gather feedback on local features that could benefit from a finer resolution?

This choice was made for pratical and technical reasons.

Tehnically, given our choice (determined via testing) to spectrally nudge the 45-km grid's large-scale temperature and horizontal winds above the boundary layer to prevent model drift, we were concerned about how any such simulated feedbacks may be obfuscated by the nudging. We did explore the option of two-way nesting however, which led to a practical limitation. Specifically, we were ultimately downscaling to 3 km (not the subject of this manuscript), and continual crashes were found at the grid interfaces (45 with 9 and 9 with 3) which in some locations bifurcated complex terrain. In short, nudging choices and grid location led us to choose a one-way nesting approach.

3. I'm uncertain about the method the authors used to determine the projected changes in temperature (K K$^{-1}$) and precipitation (mm d$^{-1}$ K$^{-1}$) per degree of global warming in Figure 7. Just to clarify, are these changes being calculated only for the end of the 21$^{st}$ century? Please let me know if I understood correctly. If so, is the calculation of "global warming" based on the ensemble mean derived from all sixteen GCMs for the entire globe?

Thank you for the opportunity to clarify. The changes you see in Fig. 7 are being computed for the 2070-2100 period (end-century) relative to the historical period (1980-2010). In the instance of precipitation changes, we compute the ensemble-mean precipitation change [mm/d] and divide by the ensemble-mean global temperature change [K]. We compute anthropogenic response this way for consistency with the IPCC and because this approach effectively eliminates scenario uncertainty. For instance, considering the CESM2 experiments only, if we compute the change in precipitation normalized by the amount of global warming for mid-century (2030-2060), we will get a similar plot to another version in which we compute the change in

precipitation for end-century normalized by the end-century global warming. We also see similar maps of of the global mean temperature-normalized precipitation response between the SSP2,3, and 5 emission trajectories of the CESM2 experiments ofr the end-century period:

[Figure]

**Non-text references**

Bass, B., Rahimi, S., Goldenson, N., Hall, A., Norris, J., and Lebow, Z. J.: Achieving Realistic Runoff in the Western United States with a Land Surface Model Forced by Dynamically Downscaled Meteorology, Journal of Hydrometeorology, 24, 269–283, https://doi.org/10.1175/JHM-D-22-0047.1, 2023.

Zobel, Z., Wang, J., Wuebbles, D. J., and Kotamarthi, V. R.: High-Resolution Dynamical Downscaling Ensemble Projections of Future Extreme Temperature Distributions for the United States, Earth's Future, 5, 1234–1251, https://doi.org/10.1002/2017EF000642, 2017.

Zobel, Z., Wang, J., Wuebbles, D. J., and Kotamarthi, V. R.: Evaluations of high-resolution dynamically downscaled ensembles over the contiguous United States, Clim Dyn, 50, 863–884, https://doi.org/10.1007/s00382-017-3645-6, 2018.

Rahimi, S., Krantz, W., Lin, Y.-H., Bass, B., Goldenson, N., Hall, A., Lebo, Z. J., and Norris, J.: Evaluation of a Reanalysis-Driven Configuration of WRF4 Over the Western United States From 1980 to 2020, Journal of Geophysical Research: Atmospheres, 127, e2021JD035699, https://doi.org/10.1029/2021JD035699, 2022.

---

## Author Comment (AC4)

**Responses to RC1: GMD-2023-162**
**Stefan Rahimi et al.**

The reviewer comments are presented followed by underlined author responses.

The authors provided an overview of the WUS-D3 dataset for the complex terrain western United States using the dynamical downscaling method based on 14 CMIP6 GCMs historical and SSP3-7.0 scenario simulations plus two projections SSP2-4.5 and SSP5-8.5 of one GCM (CESM2). The atmospheric model WRF with the horizontal resolution of 9 km and 39 vertical levels was used to conduct downscaling simulations. The authors described the challenges of producing WUS-D3 dataset, including GCM selection and technical issues, as well as an evaluation for the simulations' realism by comparing historical results to temperature and precipitation observations. They concluded that because of its high resolution, comprehensiveness, and representation of relevant physical processes, this dataset presents a unique opportunity to evaluate societally relevant future changes in western U.S. climate.

The method is unusual: "We downscale each GCM year separately and in parallel; at the beginning of each downscaling period (on August 1), the RCM is initialized to the driving GCM state."

The reason given is not yet convincing: "WRF's parallelization procedure, which is advantageous for executing simulations in weeks instead of years, is performed to the detriment of time continuity in simulating the surface and subsurface runoff with high precision." With that deficit, the soil moisture memory is neglected in these simulations. Not using a continuous simulation, will lead to unrealistic jumps in storage variables, especially in soil moisture or the snowpack. This, in turn, may cause effects in other variables, such as evapotranspiration, latent heat flux, albedo and 2m temperature. Hence, the authors should discuss the potential deficits of the dataset implied by not using a transient simulation.

This is indeed a substantial limitation of this dataset. From the outset, we committed to some type of parallelization strategy to reduce integration times similar to other studies (e.g., the previous works of Zobel et al. below). However, we were not initially sure how much spin-up time was to be used nor how to parallelize. We eventually justified a one-month spin-up in reanalysis-direven tests in Rahimi et al. (2022), Here, we conduycted two year-long test experiments for water year 2010. In case 1, we used a single month of spin-up, and in test 2, four years of spin-up were integrated. Broadly speaking, we found there to be minimal differences in simulated soil moisture, soil temperature, surface air temperature, and precipitation between the two cases.

We are wary of the spin-up issues and resulting discontinuities in land-surface variables. For example, snow in WUS-D3 simulations is generally far too wet over the historical period, a feature common to different GCMs. By the end of each simulated year (31 August), snow does not completely melt out at all locations, leading to a discontinuity in the snow fields between 31 August and 1 September. Across these areas, this results in discontinuities in surface energy fluxes as the reviewer suggests. We have thus added Sec. 2.6 to the manuscript cautioning end-users about this issue:

'       Despite one month of spin-up in parallelized yearly WRF experiments, our adopted spin-up strategy neglects high-resolution soil memory on time scales greater than one month. This assumption may be particularly problematic across regions where a transient simulation is necessary to equilibrate the soil conditions to a state which properly resolves the local-scale land-atmospheric coupling. For instance, some grid points do not see complete meltout of snow by 31 August 1993, but since data is retained from 1 September 1993 onwards, there are instances where discontinuities in surface snow coverage exist. This leads to discontinuities in surface energy variables (e.g., sensible heating; not shown). We encourage end-users of WUS-D3 to be wary of this pitfall. To alleviate this discontinuity, we propose that the atmospheric temperature, precipitation, surface radiative fluxes, winds, and specific humidity from WRF be used to drive offline calibrated hydrology models that are time-continuous and can be integrated much more rapidly (e.g., Bass et al., 2023). We acknowledge that this approach is inadequate across regions with a strong land-atmosphere coupling.

The analyses do not state much about the difference between WRF-GCMs with SSP2-4.5, 3- 7.0 and 5-8.5. How to explain in Mid-Century (Figure 6, left) that the mean temperature change of SSP2-4.5 of CESM2 (O) is greater than SST3-7.0 (A)? And why in End-Century (Figure 6, right), the SSP3-7.0 (A) has the longer arrow than O and P?

Despite large differences in greenhouse gas concentrations, the forced response of the SSP2,3 CESM2 simulations (and their downscaled simulations) tends to be maximized for the end-century (EC) period and closer during the mid-century periods. After 2050, the SSP2 and 3 scenarios diverge most prominently in terms of population growth and cropland area, Given this similarity through mid-century, it is entirely plausible that the SSP2 experiment (O) could be warmer than the SSP3 experiment (A) in light of internal variability. Specifically, A may just be in a 'cool' phase with respect to its internal variability in the 2030-2060 climate mean, while O may be in a warm phase.

As for why A (CESM2 SSP3) has a longer arrow than O or P (CESM2 SSP2 and SSP5, respectively) in the mean across the western U.S., this is tricky to answer. The arrow indicates the 'largeness' of the downscaling-induced modification of the climate change signal from the that of the parent GCM. Of course, the precipitation and warming responses become larger from SSP 2 to 3 to 5 (as expected). However, this does not mean that downscaling modifies the climate change signal in the same way between these experiments. We would contend that the modification in the western U.S. mean is small between A , O, and P, with the modifications in A (length of the arrow) constituting a drying and cooling of the original change signal by 0.4 K and 0.05 mm/d, respectively. More generally, this figure was presented to illustrate the regional modification of the change signal by downscaling. To clarify this, we have added the following sentence to the end of P1 of Sec. 4: 'The purpose of Fig. 6 is to illustrate the degree to which downscaling can modify the original GCM projections.'

Finally, we prioritized SSP3, and perhaps scenario uncertainty is a weakness of WUS-D3. However, we did take a look at the precipitation response per degree of global

warming (end-century minus to historial era) for CESM2 and found that this quantity was quite similar in our downscaled data:

[Figure]

Any outlook for bias reduction/correction? 5 K cold bias seems to be quite large. Implications of this large bias should be discussed.

So true… so, this is a whole other wing of our downscaling thrust – to ascertain best practices of bias correction in dynamical downscaling. The standard right now is for these fields to be bias corrected *after* dynamical downscaling. Even in spite of downscaling the relatively best performing GCMs, we still arrived at incredibly biased solutions. Such bias solutions are often 'covered up' by post-downscaling bias correction, and it is ubiquitous in hydrology or demand forecasting modeling. We have subsequently added the follow text to the manuscript's discussion, P3:

'The dynamical downscaling community should be frank about such biases, particularly in lieu of the fact that these biases are often artificially removed post-downscaling using bias correction. This practice is ubiquitous in hydrology and demand forecast modeling, as well as in statistical downsing. End-users of WUS-D3 should be open-eyed and wary about the possibility that these large historical biases may compromise the trustworthiness of the climate change signal.'

Discussion and outlook: as the considered area (WUS) is next to the open Pacific, has any atmosphere-ocean coupled model been applied for this region for downscaling CMIP6? What would be the role of the Pacific Ocean on regional climate over WUS, at least in the 45 km set up which cover half of ocean?

This is an interesting idea! So, SSTs, prescribed from the parent GCM, are updated every 6 hours in WRF. Since large-scale temperature and winds from the native GCMs are preserved in downscaling via spectral nudging as well, this assumes that the atmospheric-ocean coupling to a first order is also preserved in downscaling. To our knowledge, there have been no studies conducted across the region to assess a

regional ocean coupling and its subsequent effects on climate (a version of WRF des exist for this!). To that end, we have added the following to the conclusions (P5):

'Despite the care taken in creating WUS-D3, this manuscript provides a forum to scrutinize dynamical downscaling technique. For instance, here we assume that the ocean-atmosphere coupling is adequately preserved in downscaling since SSTs are prescribed to update regularly, and large-scale winds and temperatures are preserved in downscaling via spectral nudging. But, is this a good assumption given that half of our 45-km grid covers the open Pacific, so should a version of WRF with coupled ocean capabilities be used in future dfownscaling across the region? Also, as discussed previously, unrealistically large surface air temperature and precipitation biases in the parent GCMs were in some cases replaced by equally egregious biases in the downscaled solution. Despite a careful GCM selection process employed in this study, does this result motivate the consideration of a bias correction procedure for future downscaling?'

**Minor comments**

Lines 66-76: Arguments are not clear. Please rephrase the paragraph.

Sorry about this – the paragraph is a bit rambling. The intention of this paraghraph is to (i) state that dynamical downscaling of GCMs is far less common as a practice than dynamically downscaling reanalyses from a regional weather and climate modeling perspective and (ii) motivate its strengths over other downscaling methods. We have modified the paragraph to read:

'Direct dynamical downscaling of GCMs is far less common than that driven by historical reanalyses (Liu et al., 2017, 2011; Rahimi et al., 2022; Rasmussen et al., 2011, 2014; Norris et al., 2019, and many, many others) due to the fact that historical reanalyses tend to more reliably contain the requisite data to drive RCMs (Bruyère et al., 2014; Coppola et al., 2020, 2021; Huang et al., 2020, 2021; Komurcu et al., 2018; Wang and Kotamarthi, 2015, 2013; Zobel et al., 2018, 2017; Bukovsky and Karoly, 2011; Bukovsky et al., 2021; Mearns et al., 2012; Scalzitti et al., 2016). Further, since dynamical downscaling uses the laws of physics to arrive at the high-resolution end-product, it can be superior to other purely statistical-based downscaling methods. For example, dynamical downscaling does not explicitly assume stationarity (Lanzante et al., 2018) in the creation of future projections, as with other forms of downscaling (e.g., statistical); the parameterization choices whin RCM do contain empirically-derived assumptions that are not completely free of time stationarity. Dynamical downscaling can however can be used to tie explicitly simulated extreme weather events to the governing large-scale dynamics simulated within their driving GCMs. Additionally, RCMs can solve for the full complement of physical quantities relevant to climate that are otherwise not available in statistical downscaling, which typically focus on a small set of variables,. For example, statistically downscaled precipitation and temperature data products, even when obtained using multivariate relationships, may contain no information about water vapor content, surface pressure, cloud depth, etc. Finally, the

use of physics to arrive at the downscaled result means that feedbacks between the landscape and the overlying atmosphere, and other land and atmosphere processes, may be effectively simulated (e.g., the snow-albedo feedback).'

Line 92: Please list here names of the 11 states and display them on Fig.1. Not all readers are familiar with their locations.

Great suggestion. Done!

Line 106-107: "SSP-2-4.5 and SSP-5-8.5" should be "SSP2-4.5 and SSP5-8.5" to be consistent with SSP3-7.0 and themselves on other pages.

Indeed. Changed!

Line 126: Sentence is not clear: "To address this issue, we propose that the atmospheric fields from WRF be used to drive offline and calibrated hydrology models that are continuous". Please rewrite it.

This sentence has been moved to section 2.6, and we have provided additional context: 'We encourage end-users of WUS-D3 to be wary of this pitfall. To alleviate this discontinuity, we propose that the atmospheric temperature, precipitation, surface radiative fluxes, winds, and specific humidity from WRF be used to drive offline calibrated hydrology models that are time-continuous and can be integrated much more rapidly (e.g., Bass et al., 2023). We acknowledge that this approach is inadequate across regions with a strong land-atmosphere coupling.'

Line 128: Which kind of aerosols were used in WRF for these simulations? It's expected to use transient aerosols for such historical-scenario simulations.

Neither time- nor spatially-varying aerosols were used in our WRF simulations. To that end, we have included the following text in the paragraph in question, "Because coupling WRF to an atmospheric chemistry model is 6-20 times more computationally expensive, transient aerosol forcings were not considered in our study."

Line 133: Why not using the transient land-use/land-cover from CMIP6?

This is an extremely challenging item to implement in WRF and is the subject of ongoing research. As a result, this may locally impact the climate change signal. We have added the following text to make it clear which external forcings are being explicitly or implicitly considered, 'Further, historical-era 21-category land-use/land-coverage (LULC) information from the Moderate Resolution Imaging Spectrometer is used in all experiments. Since CMIP-projected LULC changes were not implemented in WUS-D3, the anthropogenic forcings considered in this study stem directly from carbon dioxide and methane concentrations, and indirectly from greenhouse gas, aerosol, and LULC forcings in the forcing GCMs at the lateral boundaries.'

Line 190: "if at all in CMIP6 GCMs": not clear what the authors mean here

We can see why this is unclear. We have deleted the 'if at all' clause of the sentence.

Line 262: What is the resolution of PRISM dataset?

~4 km grid length. We have added the resolutions in the sentence which now reads, 'We compare the downscaled ensemble mean against the native-resolution GCM ensemble mean, in addition to 9-km WRF-ERA5 and observational estimates from the 4-km Parameter-elevation Regressions on Independent Slopes Model (PRISM; Daly et al., 1994). '

Line 272: Should the "black circles" be plotted in red to increase eyes-catching effect?

Great idea! Done.

Line 272-273: Any explanation/speculation for the result of "Exceptions are noted across some western states, especially in winter"?

So, the spread is tied varying mean-state biases in parent GCMs. We currently have two papers undergoing revisions in GRL on the matter. In short, mean-state biases in GCM 3-D temperature, and, to a lesser degree, winds and SSTs are correlated with how biased the dynamically downscaled precipitation and temperature are. We have thus added the following text to the manuscript, 'Exceptions are noted across western states, especially in winter; we speculate that dynamical downscaling is increasing the spread proportional to the magnitude of GCM bias in temperature, winds, and SSTs which, when inherited by WRF, leads to varying magnitudes of downscaled precipitation and temperature bias. GCM bias impacts on the dynamically downscaled solution are a current core focus by our research team.'

Line 277-278: What does "meaningful subregional biases of hundreds of percent" mean here?

Since we have included state labels in Fig. 1, I have made the sentence more specific, 'These biases vary substantially within the ensemble, with individual downscaled GCMs exhibiting meaningful state-wide biases of hundreds of percent (e.g., California in May for CNRM-ESM2-1; not shown).'

Line 290-291: Sentence is not clear

We have rewritten the text here for clarity, 'In general, overly wet and cold dynamically downscaled GCMs have previously been noted across the region with a different RCM (Rastogi et al., 2022), indicating that biases in the GCM forcing data may be to blame. The effects of GCM bias propagation are being explored in Rahimi et al., (2023; in revisions) and Risser et al. (2023; in revisions). The absence of such large biases in

WRF-ERA5 (Figs. 3, 4, and 5), which is equivalent to the downscaled GCMs, except driven by ERA5, lends further evidence in support of this hypothesis.'

 Line 292: How was "rx1day" defined/determined?

At a gridcell, we take precipitation on the wettest day of the year and average this over a climate period (e.g., 1980-2010) and is a popular metric to gauge extreme precipitation. We have not made any alterations to the text for further clarification.

Line 324-325: "Despite a positive mean change, a handful of simulations suggest drying across the region.": where does the information come from?

This information comes from Fig. 6, right (added to main text), in which simulations B, C, N, and L predict a negative mean precipitation change by end-century.

Line 326: "For warming amounts,..."? Should it be "For temperature change,..." as on the line 328, the "precipitation change" is mentioned.

Done.

Line 341: Typo on title of section 4.1 "Patters" instead of "Patterns"

Thank you! Fixed.

Line 385: What are "ensemble-mean fractional precipitation changes"?

Ah yes, we are referring the precipitation changes relative to the historical era. We have added a clarifying sentence, 'Here, a value of -20% K$^{-1}$ indicates that EC-era precipitation has decreased by 20% relative to the historical-era while the global temperature has warmed by 1 K.'

More generally, we are using the %/K framework to align our messaging with the Clausius-Clapeyron equation, in which atmoppheric water vapor content increases at 7%/K.

Line 433-434: Sentence is unclear

Ah yes, this is tricky to communicate. So, the idea is that we first compute the 'actual' future changes inTmax99 exceedence days in the dynamically downscaled ensemble in the left panel. We then compute the climate-mean changes in Tmax99 between the future and historical eras, and we add this delta  Tmax99 to the historical Tmax99 and repeat the calulation to obtain the change in exceedance days in what we are calling a 'mean shift' assumption (center panel). The point of this comparison is to show that we are getting lengthening in the tails of the temperature distribution that cannot be explained by mean shifts alone. This indicates that the most extreme events are warming more across parts of the domain relative to cooler events.

For clarity, we have made heavy edits to this paragraph which now reads, 'Next, we explore whether changes in Tmax99 exceedances are explainable by mean shifts in the temperature distribution. As shown in Fig. 10 (right column), the number of actual Tmax99 exceedances from parent and dynamically downscaled GCMs can be quite different compared to the case where all quantiles in the temperature distribution are shifted equally based on the amount of local mean warming in Tmax99 (Fig. 10; middle).  Red (blue) pixels indicate regions where the tails of the temperature distribution are warming more (less) than can be explained by mean warming in Tmax99. Assuming a mean shift in Tmax99 significantly underpredicts the increase in exceedances by 3-4 days per year $K^{-1}$ across portions of California, Oregon, and Washington. Still greater underpredictions of future exceedances assuming a mean shift are seen across western Montana, Idaho, and portions of western Wyoming, particularly at higher elevations. Further south however, the number of exceedances in Tmax99 can be explained mostly by mean shifts in Tmax99. Assuming a mean shift, exceedances are slightly overpredicted across portions of New Mexico and western Texas relative to GCM and WRF simulations. This analysis highlights that the intensification of extreme temperature events may not be entirely explainable by mean shifts in the temperature distribution alone, and parent and downscaled GCMs are broadly similar in this respect. However, there is also significant spatial structure in the downscaling patterns not seen in the GCMs, indicating that local atmospheric dynamics and local land-atmosphere feedbacks play a role in shaping change in the right tail of the temperature distribution.'

Line 434: Should it be (Fig.10, middle) instead of (Fig.9, middle)?

Yes, done!

Figure 3 (and some elsewhere): It would be easier to compare if the PRISM dataset figure and the WRF-ERA5 figure locations are exchanged

Done!

Figure 8 and Figure S9 caption: Should "Stippling is not included for temperature because every grid point returns a p value smaller than 0.05." be removed?

At the current time, we have chosen to leave this in to convey that shifts in temperature are statistically significant.

Figure 8 (line 414): There is no a, b, c, d on the Figure 8

For Fig.8, we have removed a, b, c, and d, and have opted for descriptive titles and words that reference subpanels.

Figure S2 caption: Should move the unit [K] after the word "biases" like this: "11-state-mean biases [K] are presented beneath each GCM label."

Done!

Figure S5 caption: Should add the unit [K] after "annual-mean surface air temperature"

Done!

Figure S7 caption: Typo in "the" 16-GCM mean. Should remove "1 April" as the figure shows three months (Jan Apr, Jul)

Done!

Figure S10 caption: Should move [mm d⁻1] after "rx1day precipitation"

Done!

Figure S11 caption: [% K⁻1] should be located after "rx1day precipitation"

Done!

**Non-text references**

Bass, B., Rahimi, S., Goldenson, N., Hall, A., Norris, J., and Lebow, Z. J.: Achieving Realistic Runoff in the Western United States with a Land Surface Model Forced by Dynamically Downscaled Meteorology, Journal of Hydrometeorology, 24, 269–283, https://doi.org/10.1175/JHM-D-22-0047.1, 2023.

Zobel, Z., Wang, J., Wuebbles, D. J., and Kotamarthi, V. R.: High-Resolution Dynamical Downscaling Ensemble Projections of Future Extreme Temperature Distributions for the United States, Earth's Future, 5, 1234–1251, https://doi.org/10.1002/2017EF000642, 2017.

Zobel, Z., Wang, J., Wuebbles, D. J., and Kotamarthi, V. R.: Evaluations of high-resolution dynamically downscaled ensembles over the contiguous United States, Clim Dyn, 50, 863–884, https://doi.org/10.1007/s00382-017-3645-6, 2018.

Rahimi, S., Krantz, W., Lin, Y.-H., Bass, B., Goldenson, N., Hall, A., Lebo, Z. J., and Norris, J.: Evaluation of a Reanalysis-Driven Configuration of WRF4 Over the Western United States From 1980 to 2020, Journal of Geophysical Research: Atmospheres, 127, e2021JD035699, https://doi.org/10.1029/2021JD035699, 2022.

---

## Author Comment (AC5)

**Responses to CC1: GMD-2023-162**
Stefan Rahimi et al.

The reviewer comments are presented followed by underlined author responses.

The paper presents a comprehensive study introducing downscaling work to a 9 km resolution for 16 CMIP6 GCM experiments using the WRF model. The manuscript well describes the methodology and the WUS-D3 dataset. I recommend publishing the paper after some minor revisions as follows.

Comments:
1. GCM selection: the authors outlined 6 processes considered in the evaluation and selection of GCMs. While they refer to the ranking methodology in a technical note (Krantz et al. 2021) and a paper currently under revision (Goldenson et al. 2023, in revisions), I recommend providing more information on two key aspects. Firstly, elaborate on the process selection – explain why these 6 processes were chosen; why extreme precipitation across California is included among the selected processes, given the coarse resolution of GCMs and the fact that this diagnostic variable might not play a role in the ICBC of the downscaling framework. Secondly, provide more details on the ranking methodology: clarify how these 6 processes are considered in the final ranking; are they equally weighted? How do temporal and spatial patterns contribute to the selection process?

Happy to provide additional context and clarification here. For the first question, the processes were chosen based on our team's experience in processes that are of regional relevance. For California specifically, we actually considered a set of metrics for biases within just this category. Specifically, we considered the following, conditioned on days when GCM precipitation exceeded the 95th percentile (biases relative to ERA5): integrated water vapor, sea level pressure, and the 250 hPa zonal wind. We also used the third empirical orthogonal function (EOF) of the 500 hPa geopotential, whose spatial pattern is strongly correlated with extreme precipitation (> 99th percentile) across the region (See Chen et al., 2021). Additionally, we also included a metric for large-scale circulations from the GCM that may affect Los Angeles extreme precipitation.

To answer the first question, the metrics of GCM bias were of hemispheric, Pacific Ocean, western U.S., and California scope. Since California is so latitudinally expansive however, and because the landfalling atmospheric rivers that bring the state extreme precipitation generally provide abundant rainfall downstream to the western U.S. interior, we believe that including biases across California should be regarded as regional versus point biases. Further, since the large-scale patterns of temperature and horizontal winds are preserved above the boundary layer via spectral nudging on the 45-km WRF grid, we believe that consideration of biases in the large-scale dynamic fields associated with extreme precipitation events across California, have regional-scale and western U.S. consequences.

Regarding the second question, spatial patterns and temporal patterns are considered in the selection process. For instance, the time-variability of ENSO and high-frequency

synoptic variability of landfalling waves are considered, while the spatial variability of the California precipitation mode is considered via the identification of where the geopotential anomalies exist upstream of the region. Additionally, our metrics per Simpson et al., (2020) consider jet stream landfall position bias, accounting for spatial bias. More generally however, these processes were not considered equally in the finalized GCM selection process. First, metric redundancy was addressed by computing a set of EOFs from the metrics, and only retaining a subset of EOFs that capture most of the variation between models. The result is a reduced set of linear combinations of metrics that efficiently captures nearly all of the variance across GCMs; this process constituted a weighting of the metrics themselves based on redundancy with other metrics. It was found that the first 6 EOFs described 91% of the variance amongst models, with only a subset of metrics explaining most of the variability between GCMs. After the EOF decomposition, and overall score was computed for each GCM. The least biased (highest scoring) GCMs were then generally selected.

We have rewritten Sec. 2.2 to read: 'Prioritizing SSP3-7.0 with an end-of-century radiative forcing of 7 W m$^{-2}$, we selected 14 GCMs (Table 1) based on three criteria: (i) their skill in simulating important processes that govern western North American climate over the historical (1980-2010) period, (ii) their collective representativeness of the broader CMIP6 ensemble spread in future temperature and precipitation responses, and (iii) data availability. Aspects considered in the GCM evaluation included:

1. Large-scale meteorology associated with Santa Ana and Diablo winds – important for extreme wind and fire risk across the southwestern U.S. We use this metric to minimize the usage of GCMs which simulate a distorted portrayal of the Pacific High.
2. The El Niño Southern Oscillation (ENSO) – well-known to modulate the interannual variability of precipitation and temperature across the western U.S. We use this metric to prioritize GCMs which adequately capture the ENSO-Western U.S. teleconnection.
3. Northern Hemisphere blocking and circulation (Simpson et al., 2020) – Wave characteristics, both over climate and synoptic time scales, are directly related to the variability of precipitation across the Western U.S. We use this metric, for instance, to ensure that GCMs are down-selected if they are too progressive in their simulation of mid-latitude waves.
4. Landfalling jet characteristics – Atmospheric rivers are responsible for a majority of West-Coast precipitation. As such, we only select GCMs that demonstrate superior performance in their landfalling position and tilt.
5. GCM-simulated surface air temperature and precipitation – while these variables can be incorrectly simulated in GCMs despite the more-or-less correct treatment of their local driving processes, which may be more important for driving a regional climate model, we include these variables to account for the relationships between the GCM-simulated processes and GCM-simulated surface temperature/precipitation profiles.
6. Extreme precipitation across California – Generally, extreme precipitation events in California are driven by large-scale synoptic events (described by column water vapor, 500 hPa geopotential, and upper tropospheric wind speeds). These

large-scale patterns can have ramifications for weather and climate as they propagate downstream, hence we include an evaluation of bias in these fields for our GCM selection.

7. Regional wind shear – Wind shear helps to moduleate the lifetime of precipitation systems through storm-scale organization and is a measure for the larger-scale background baroclinicity which is important for storm tracks. We thus evaluate its bias.

The ranking system is described in Krantz et al. (2021), and the process of choosing GCMs to downscale based on end-user needs and locally relevant atmospheric processes is described in Goldenson et al. (2023). To emphasize, being subject to these selection processes, the GCMs downscaled in this study span the range of future changes in temperature and precipitation from CMIP6 across the WUS.

For more details on the GCM selection process, we refer readers to Krantz et al., (2021). However, we highlight that temporal and spatial variability was considered in ranking a preferred set of GCMs to downscale. Specifically, the time-variability of ENSO and high-frequency synoptic variability of landfalling waves are considered, while the spatial variability of the California precipitation mode Chen et al. (2021) was factored into our analyses via the identification of where the geopotential anomalies exist upstream of the western U.S. on extreme precipitation days. Additionally, our metrics per Simpson et al., (2020) consider jet stream landfall position bias. Finally, Krantz et al. (2021) performed a variance decomposition using empirical orthogonal functions to reduce the effects of metric redundancy, weighting them accordingly in the final rankings of GCMs.'

2. L330: The authors stated that "Interestingly, downscaling generally reduces warming (leftward pointing arrows)" and hypothetically attributed it to the reduced snow albedo feedback with downscaling. I recommend that the authors prove this hypothesis by comparing the snow outputs of both WRF and GCMs.

The suggestion of a stronger warming response in the GCMs relative to WRF is intended only as a hypothesis and thus needs to be tested. Our group plans to look at this in another paper (in preparation).

3. The authors conducted a more in-depth analysis of the changes in rx1day and tmax99. However, there is no explanation as to why only these two indices, among many possible extreme indices, were selected. Furthermore, why did the authors opt for the absolute index (rx1day) when analyzing rainfall, while choosing the percentile index for temperature.

The purpose of this manuscript was to present the dataset rather than conduct extensive scientific process studies and analyses of extremes. However, we wanted to inspire the community to use the dataset, so we conducted initial analyses to examine mean changes in mean temperature and precipitation, as well as rx1day precipitation and Tmax99. We looked at these common extreme metrics (and mean changes) to showcase how the footprint of topography is represented in the climate response, a feature not characteristic of the GCMs. Rx1day is a common metric for extreme

precipitation while Tmax99 is also common in extreme heat analyses, so for an overview of the dataset, we thought that presenting these two metrics alone would be enough to inspire community analysis. We certainly acknowledge that this analysis was by no means comprehensive, and future studies using WUS-D3 should include a more expansive set of metrics.

Minor comments:
1. L210, 215 should refer to Table 1's last column. The caption of Table 1 should also provide an explanation of the last column (SST mode)

Done!

2. Please add the names of locations mentioned in the text to Figure 1, such as California's Central Valley, Sierra Nevada, and state names, …

This is a great idea and has been done for state names. Regarding the mountain ranges, this may be difficult since we list 5-6 over a large geographic region. Indicating these regions in Fig. 1 may clutter the figure. Thus for now, we only include state names.

**References**

Krantz, W., Pierce, D., Goldenson, N., and Cayan, D.: Memorandum on Evaluating Global Climate Models for Studying Regional Climate Change in California, *The California Energy Commision,* https://www.energy.ca.gov/sites/default/files/2022-09/20220907_CDAWG_MemoEvaluating_GCMs_EPC-20-006_Nov2021-ADA.pdf, 2021.

Simpson, I. R., Bacmeister, J., Neale, R. B., Hannay, C., Gettelman, A., Garcia, R. R., Lauritzen, P. H., Marsh, D. R., Mills, M. J., Medeiros, B., and Richter, J. H.: An Evaluation of the Large-Scale Atmospheric Circulation and Its Variability in CESM2 and Other CMIP Models, Journal of Geophysical Research: Atmospheres, 125, e2020JD032835, https://doi.org/10.1029/2020JD032835, 2020.

---

## Author Response (AR2)

**Responses to reviewers (2):** GMD-2023-162
Stefan Rahimi et al.

Reviewer comments are presented with underlined responses.

**Referee 1**

L166-167: "Because coupling WRF to an atmospheric chemistry model is 6-20 times more computationally expensive, transient aerosol forcings were not explicitly considered in our study." Here should be "interactive aerosols" if coupling WRF to an atmospheric chemistry model is mentioned. The transient aerosol forcing can be taken from CMIP6 or other sources. The authors should clarify whether the aerosol climatology is used in the simulations or not. If not, what else?

Reply: Changed 'transient' to 'interactive'. The transient aerosol forcings and their effects on dynamic variables are contained within the GCM boundary condition data

L309: typo at "absence"

Reply: I could not find this typo at this location but corrected the same typo at the end of Sec. 2.3. I think there might be a discrepancy in the line numbering between the reviewers' versioning of the revised manuscript.

L613: should be numbered as Section 5 (not 7)

Reply: I cannot find this error at line 613 but have checked to make sure that it was removed at line 516.

**Referee 2**

No comments to address